# Factored Latent Action World Models

Zizhao Wang [* 1]   Chang Shi [* 1]   Jiaheng Hu [1]   Kevin Rohling [1]
Roberto Martín-Martín [1]   Amy Zhang [1]   Peter Stone [1 2]

## Abstract

Learning latent actions from action-free video has emerged as a powerful paradigm for scaling up controllable world model learning. Latent actions provide a natural interface for users to iteratively generate and manipulate videos. However, most existing approaches rely on monolithic inverse and forward dynamics models that learn a single latent action to control the entire scene, and therefore struggle in complex environments where multiple entities act simultaneously. This paper introduces Factored Latent Action Model (FLAM), a factored dynamics framework that decomposes the scene into independent factors, each inferring its own latent action and predicting its own next-step factor value. This factorized structure enables more accurate modeling of complex multi-entity dynamics and improves video generation quality in action-free video settings compared to monolithic models. Based on experiments on both simulation and real-world multi-entity datasets, we find that FLAM outperforms prior work in prediction accuracy and representation quality, and facilitates downstream policy learning, demonstrating the benefits of factorized latent action models.

## 1. Introduction

Recent advances in Latent Action Models (LAM) (Schmidt & Jiang, 2023; Bruce et al., 2024) have unlocked the possibilities of learning world models from action-free videos that are abundant on the web. Specifically, these approaches use an inverse dynamics model to encode environmental changes into a *single* latent action. The latent action is then used to train a forward dynamics model, allowing controllable predictions of future frames from in-the-wild videos.

However, in-the-wild videos often contain complex scenes where many entities may be taking actions simultaneously: for instance, a robot video may include independent arm movements and shifting camera perspectives; while a soccer game involves several players, the ball, and even background audience motion, each of which acts independently. Compressing all these motions into a *single* latent action is challenging, since the complexity of the underlying action space grows exponentially with the number of movable entities (Fig. 1). Consequently, existing methods struggle with latent action learning in such settings, which severely limits their applicability to diverse scenarios.

This paper introduces the Factored Latent Action Model (FLAM), where the latent state is decomposed into a set of factors, each independently predicting its latent action and its next-step value via shared factored inverse and forward dynamics models. Compared to prior work that must capture all joint action combinations within a single latent action space, FLAM assumes a shared latent action space across factors and thereby reduces the learning problem to modeling each entity's action patterns within this common space. Additionally, regarding forward dynamics, unlike most prior LAM approaches that use a monolithic scene representation entangling all entities, FLAM factorizes the scene into compositional entities with a shared forward dynamics model, inherently supporting permutation invariance and enabling stronger generalization. With next frame prediction as the training objective, FLAM learns factorized state and action representations from action-free video data, leading to more accurate modeling of complex, multi-entity dynamics and improved prediction quality compared to previous work.

Based on experiments on both simulated and real-world multi-entity datasets, including autonomous driving videos, we find that FLAM outperforms prior state-of-the-art methods in both prediction quality and factor-entity correspondence. Moreover, the inferred latent actions capture essential entity behaviors and enable sample-efficient policy learning. Qualitative results are available at https://sites.google.com/view/factored-lam.

## 2. Related Work

In this section, we first discuss previous works in learning world models from action-free videos. Next, we dis-

---

[*]Equal contribution  [1]University of Texas at Austin [2]Sony AI. Correspondence to: Zizhao Wang <zizhao.wang@utexas.edu>.

*Proceedings of the 43rd International Conference on Machine Learning*, Seoul, South Korea. PMLR 306, 2026. Copyright 2026 by the author(s).

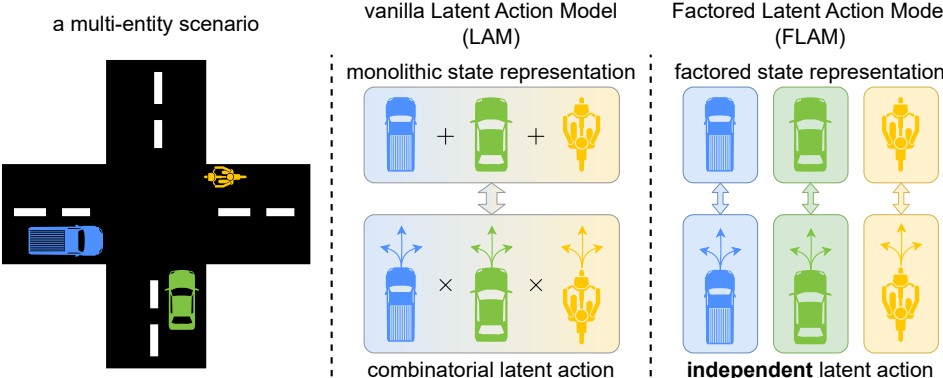

*Figure 1.* In multi-entity scenarios, **(left)** such as an intersection with three road users: **(middle)** a vanilla latent action model encodes the scene change with a *single* latent action of dimension $d$, which makes learning challenging as this latent action space needs to model all $|\mathcal{A}|^K$ joint action combinations. **(right)** In contrast, FLAM decomposes the state into $K$ factors, each with its own latent action of dimension $\frac{d}{K}$. Additionally, we assume all latent actions share the same space (i.e., with the same prior / codebook), which reduces the learning problem to modeling the $|\mathcal{A}|$ actions per factor rather than their $|\mathcal{A}|^K$ joint combinations.

cuss prior works in factorized decision making and object-centric representations, whose strength and weakness inspire FLAM.

**Dynamics Modeling without Action Labels**. Given the abundant source of videos and the scarcity of action labels, several methods have been developed to learn from pure observations. ILPO (Edwards et al., 2019) learns a latent policy with forward dynamics model only, and later maps the latent policy output to real action through an action remapping network. LAPO (Schmidt & Jiang, 2023) and Genie (Bruce et al., 2024) jointly learn an inverse dynamics model jointly with a forward dynamics model. Nikulin et al. (2025) then introduce a small portion of ground-truth actions as dynamics modeling supervision. Ye et al. (2024) expand learning from observation from vision only to vision and language modalities. Past work (Zhang et al., 2022; Ye et al., 2022; Baker et al., 2022; Ye et al., 2024) has proven the value of dynamics modeling without action labels in applications such as learning to drive, play games, and manipulate with robot arms from videos. However, many domains include multiple entities with independent actions, and prior work falls short in modeling such complex action combinations, motivating our work to address this challenge through factorization.

**Factorization in Decision Making**. Factorization has long been used to exploit structured state and action spaces in complex environments, often through the factored MDP formulation (Osband & Van Roy, 2014; Guestrin et al., 2003). Recent works have applied this principle to derive factored forward dynamics (Pitis et al., 2020; Wang et al., 2022), factored policies (Hu et al., 2025; 2024), and factored value functions (Sodhani et al., 2022). FLAM is motivated by the same principle that factorization simplifies complexity into manageable components. However, it applies this idea to learning world models from action-free video, where the

underlying factorization is not given. Compared to prior work (Klepach et al., 2025; Daniel et al., 2026) that also focuses on world models from action-free video, FLAM is the first factored latent world model that does not use pre-trained object-centric representation and specifically focus on multi-entity scenarios.

**Object-Centric Representation Learning**. Object-centric learning aims to represent complex scenes by isolating individual objects from the background and from each other, leading to improved generalization and modeling capabilities. Key challenges include the need for supervision, leading to the development of many unsupervised methods. MONet (Burgess et al., 2019) and IODINE (Greff et al., 2019) achieved unsupervised multi-object segmentation through attention and iterative amortized inference. Slot Attention (Locatello et al., 2020a) uses iteratively updated slots to learn disentangled, object-based representations. Recent work has focused on improving object-centric learning fidelity in noisy real-world data (Jiang et al., 2023; Wang et al., 2025; Qi et al., 2025). In our work, we use Slot Attention (Locatello et al., 2020b) as the factorizer for the state representation and use the forward prediction task as the learning signal. In contrast to object-centric representation work that separates entities based on visual similarity, our representation separates entities based on the independence of their actions, yielding representations that are action-driven.

## 3. Preliminaries

Given a video dataset, we aim to model dynamics from observations alone, without any action labels. A Latent Action Model (LAM) learns an inverse dynamics model (IDM) to infer latent actions and a forward dynamics model (FDM) to predict the next observation. The inferred latent

action captures the most essential changes during a transition that cannot be predicted from the current observation alone, and thus usually correspond to entities' underlying actions. As a result, although they may not align exactly with real, physically meaningful actions, these latent actions can still provide useful information for downstream policy learning.

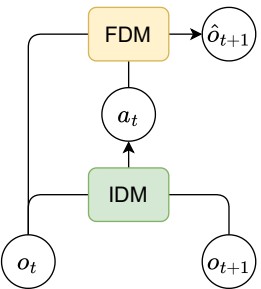

*Figure 2.* Latent action model.

As shown in Fig. 2, both the IDM and the FDM observe $o_t$, but only the IDM observes $o_{t+1}$. Therefore, to accurately predict $o_{t+1}$, the IDM must extract useful transition information through the latent action $a_t$. However, without an explicit bottleneck on $a_t$, the model can collapse to a trivial solution where $a_t$ simply copies $o_{t+1}$. To prevent this shortcut, prior work typically constrains the capacity of $a_t$, for example by discretizing it with vector quantization into a small set of codes or by regularizing it toward a prior using a VAE-style objective (Kingma & Welling, 2013; Van Den Oord et al., 2017).

While latent action models provide a general framework for learning controllable world models from videos alone, they typically use a single latent action to represent the change of an entire scene, which becomes restrictive when multiple entities act independently. For example, in crowded intersections where many cars and pedestrians are driven by independent actions, learning to compress all entities' actions into one latent action can be challenging. To address this limitation, FLAM decomposes both the state representation and the latent action into independent factors, enabling efficient modeling of multi-entity dynamics, as discussed in Sec. 4.

# 4. Factored Latent Action Model (FLAM)

From a high-level perspective, FLAM enables efficient latent action model learning in multi-entity scenarios by inferring a set of factorized latent actions rather than a single latent action between each pair of frames, via the two learning phases shown in Fig. 3 and Alg. 1:

- **Encoder learning** (Sec. 4.1): FLAM pre-trains a VQ-VAE to extract high-level features from pixels, allowing the latent action model to learn in the feature space rather than the pixel space for the purpose of efficient learning.

- **Latent action model (LAM) learning** (Sec. 4.2): Using the extracted features, FLAM decomposes the scene into several independent factors, also referred to as *slots*. For each slot, an inverse dynamics model infers a separate latent action from its current and next-frame values. Then, based on its current value and the corresponding latent action, a forward dynamics model independently predicts the next-frame value for each slot. Finally, all predicted slots are mapped back to the feature space and decoded into the next video frame.

After FLAM learns to model the world from action-free videos, its inferred latent actions can be used for either controllable video generation or policy learning to solve downstream tasks. We refer to this utilization as the third phase, and discuss how to leverage FLAM for both settings in Sec. 4.3.

## 4.1. Pretrained Encoder

As shown in Fig. 3 (a), FLAM learns a VQ-VAE (Van Den Oord et al., 2017) to extract features from raw pixels, enabling fast LAM learning. For each frame $o \in \mathbb{R}^{H \times W \times 3}$, a CNN encoder first extracts $N$ patch-level features $z \in \mathbb{R}^{N \times d_z}$. The features are then quantized from continuous encoder outputs to the nearest entry in a discrete codebook, denoted as $z_q \in \mathbb{R}^{N \times d_z}$, using finite scalar quantization FSQ (Mentzer et al., 2023), which forces features into a finite set of learned embeddings. Finally, a decoder reconstructs the frame from the quantized feature:

$$z = \texttt{Encoder}(o),$$
$$z_q = \texttt{FSQ}(z),$$
$$\hat{o} = \texttt{Decoder}(z_q).$$

The VQ-VAE is trained by minimizing the following image reconstruction loss:

$$\mathcal{L}_{\texttt{VQ-VAE}}(o) = ||o - \hat{o}||^2. \tag{1}$$

## 4.2. Factored Latent Action Model (FLAM)

As shown in Fig. 3 (b), our latent action model contains four key components: 1) a factorizer that decomposes the scene $z$ into a set of independent slots $s$, 2) a shared inverse dynamics model that infers a separate latent action $a^i$ for each slot, 3) a shared forward dynamics model that, given the current slot value and latent action, predicts the next-frame value for each slot, and 4) an aggregator that maps the predicted slots back to the feature space. These four components are jointly trained to minimize the next frame prediction error.

**Factorizer**. To decompose the scene into a set of factors with independent actions, FLAM uses slot attention (Locatello et al., 2020a). For each frame, $K$ slots $s_t \in \mathbb{R}^{K \times d_s}$

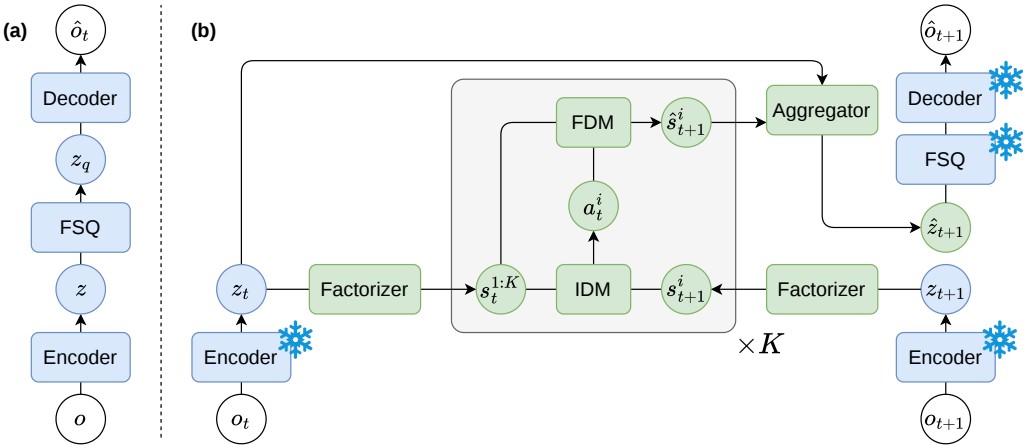

*Figure 3.* Two training stages of FLAM. **(a)** A VQ-VAE is pretrained to extract features for latent action model learning. **(b)** FLAM infers latent actions and makes predictions for each factor independently, with all modules trained jointly to minimize the prediction error.

are initialized from learned embeddings and then compete to bind to different regions in the frame through iterative slot attention. Meanwhile, to ensure that each slot consistently binds to the same object, we add an additional causal temporal attention layer so that each slot can refer to its values at all previous timestamps. Overall, the factorizer computes the slots as follows:

for each slot attention iteration

$$s_t = \texttt{Slot-Attn}(\texttt{query} = s_t, \texttt{key} = z_t), \quad (2)$$
$$s_t^i = \texttt{Self-Attn}(\texttt{query} = s_t^i, \texttt{key} = s_{1:t}^i). \quad (3)$$

**Inverse dynamics model (IDM)**. After extracting independent slots, the IDM aims to infer a latent action $a_t^i$ for each slot $i$, based on all current slots $s_t^{1:K}$ and its next-frame value $s_{t+1}^i$:

$$a_t^i = \texttt{IDM}(s_t^{1:K}, s_{t+1}^i), \quad (4)$$

where for a variable $x$, $x^{1:K}$ denotes the set $\{x^i\}_{i=1}^K$, and we use $s_t^{1:K}$ and $s_t$ interchangeably. We use all current slots rather than just $s_t^i$ as inputs to account for interactions between factors, enabling more accurate latent action prediction (e.g., a person in a car moves because of the car rather than by themselves).

To implement the IDM, we adopt the spatio-temporal model introduced in Genie (Bruce et al., 2024). The spatial block applies self-attention to capture interaction information across $s_t^{1:K}$, while the temporal block applies cross-attention to compare $s_{t+1}^i$ with its current value and encode the most meaningful changes between them:

$$\tilde{s}_t^i = \texttt{Self-Attn}(\texttt{query} = s_t^i, \texttt{key} = s_t^{1:K}),$$
$$u_t^i = \texttt{Cross-Attn}(\texttt{query} = s_{t+1}^i, \texttt{key} = [\tilde{s}_t^i, s_{t+1}^i]).$$

Note that although we use only the current and next values in the temporal block, the model can be easily extended to

condition on the full history $s_{1:t}^i$ or a fixed window $s_{t-w:t}^i$, where $w$ denotes the window length.

Similar to prior LAM methods, we regularize the capacity of the latent action to prevent it from simply copying the next state $s_{t+1}^i$ and thereby bypassing dynamics learning. Concretely, following the variational autoencoder (VAE) framework (Kingma & Welling, 2013), we project $u_t^i$ to a normal posterior distribution $q(a_t^i)$ and impose a KL divergence regularization toward a unit normal prior $p(a_t^i)$. The latent action is then sampled from this posterior as:

$$a_t^i \sim q(a_t^i) = \mathcal{N}\left(\mu(u_t^i), \sigma(u_t^i)\right),$$

where $\mu$ and $\sigma$ are the projection networks that map $u_t^i$ to the mean and standard deviation of the posterior distribution.

**Forward dynamics model (FDM)**. To provide the learning signals for the IDM, the forward dynamics model takes all current slots $s_t^{1:K}$ together with the latent action $a_t^i$ and predicts the next-frame value for each slot $\hat{s}_{t+1}^i$:

$$\hat{s}_{t+1}^i = \texttt{FDM}(s_t^{1:K}, a_t^i). \quad (5)$$

Similarly to the IDM, the same FDM is shared across all slots, and we use all current slots instead of just $s_t^i$ as inputs to capture interactions between slots, as they are not intended to be encoded by the latent action. For implementation, we use the same spatio-temporal model as the IDM, except that the temporal attention uses different queries and keys:

$$\tilde{s}_t^i = \texttt{Self-Attn}(\texttt{query} = s_t^i, \texttt{key} = s_t^{1:K}),$$
$$\hat{s}_{t+1}^i = \texttt{Cross-Attn}(\texttt{query} = s_t^i, \texttt{key} = [\tilde{s}_t^i, a_t^i]).$$

**Aggregator**. Finally, the aggregator maps the predicted slots back into the feature space, enabling decoding into the next-frame prediction. Instead of relying solely on $\hat{s}_{t+1}^i$ to predict

**Algorithm 1** FLAM

Prepare a video dataset of $(o_{1:T})$.
Initialize the VQ-VAE (`Encoder`, `FSQ`, `Decoder`) and the latent action model (`Factorizer`, `IDM`, `FDM`, `Aggregator`).
Pretrain the VQ-VAE with the reconstruction loss in Eq. (1).
// Train the latent action model
Extract features with the encoder:
$$z_{1:T} = \texttt{Encoder}(o_{1:T}).$$
Extract slots using Eq. (2) - (3):
$$s_{1:T}^{1:K} = \texttt{Factorizer}(z_{1:T}).$$
**for** slot $i = 1, \dots, K$ and time $t = 1, \dots, T-1$ **do**
  Infer the latent action: $a_t^i = \texttt{IDM}(s_t^{1:K}, s_{t+1}^i)$.
  Predict the next-frame slot: $\hat{s}_{t+1}^i = \texttt{FDM}(s_t^{1:K}, a_t^i)$.
**end for**
Map predicted slots back to the feature space:
$$\hat{z}_{2:T} = \texttt{Aggregator}(z_{1:T-1}, \hat{s}_{2:T}^{1:K}).$$
Optimize the latent action model with the prediction loss in Eq. (6).

---

the feature, the aggregator also conditions on the current feature $z_t$. This design allows $\hat{s}_{t+1}^i$ to focus on capturing changes between current and next time steps, rather than redundantly encoding static visual details. Concretely, the aggregator uses cross-attention to update each patch feature based on the predicted slots:

$$\hat{z}_{t+1} = \texttt{Cross-Attn}(\texttt{query} = z_t, \texttt{key} = \hat{s}_{t+1}^{1:K}).$$

**Training objective**. The four components of the latent action model (Factorizer, IDM, FDM, and Aggregator) are trained jointly to minimize the feature prediction error and the KL regularization:

$$\mathcal{L}_{\text{LAM}}(z_t, z_{t+1}) = \|z_{t+1} - \hat{z}_{t+1}\|_2^2 + \beta \cdot \sum_{i=1}^K D_{KL}[q(a_t^i) \| p(a_t^i)], \quad (6)$$

where $\beta$ is the KL regularization coefficient.

Note that although FLAM adopts components similar to prior work (Villar-Corrales & Behnke, 2025; Klepach et al., 2025), such as slot attention and latent action model, it fundamentally differs in its training paradigm and the resulting inductive bias. Prior work typically follows a two-stage procedure: object-centric representations are first learned in isolation, and a latent action model is then trained on top of these fixed representations. As a result, the learned slots largely mirror conventional object-centric representations and are primarily organized according to visual appearance. In contrast, FLAM jointly optimizes slot extraction and latent action model with the prediction loss, with each slot

endowed with an independent latent action in both the IDM and FDM. This end-to-end factorized dynamics learning encourages the model to separate entities based on the independence of their dynamics rather than visual similarity, yielding representations that are action-driven.

### 4.3. Learned Latent Actions Utilization

The latent actions learned by FLAM implicitly capture the controllable dynamics of individual entities in the environment, providing a natural interface for manipulating video generation. In particular, users can specify latent actions by sampling or selecting values from the prior distribution and use them as control variables during generation, resulting in diverse future trajectories.

In addition to generation, the learned latent actions encode rich information about agent behavior and enable efficient policy learning directly from video. Using a small demonstration dataset with action labels, we first train an action decoder $f$ via supervised learning to map latent actions $a$ to real actions. For larger datasets that contain only videos without action annotations, we extract latent actions using FLAM and apply the learned decoder to generate pseudo-labels for the real actions. These inferred action labels are then used to train a behavior cloning policy:

$$\mathcal{L}_\pi = \mathbb{E}_{(o,a) \sim D_E} \|\pi(o) - f(a)\|. \quad (7)$$

## 5. Experiments

Our central hypothesis is that the proposed factorized state representation and latent action formulation can better capture the features and dynamics of each entity, leading to more accurate world modeling on multi-entity videos. When used for policy learning, these improved world models should in turn yield higher downstream performance. To test this hypothesis, we evaluate FLAM on both simulation and real-world datasets and provide empirical evaluations pertaining to the following questions that support the indicated answers.

- **Q1)** Can FLAM learn more accurate world models than baselines? **Yes** (Section 5.1).

- **Q2)** Does FLAM learn action-driven factorization, grouping entities that share the same action into the same factor while separating entities with different actions? **Yes** (Section 5.2).

- **Q3)** Do latent actions extracted by FLAM enable more sample-efficient policy learning than learning without them? **Yes** (Section 5.3).

- **Q4)** How do architecture designs and hyperparameters affect performance? (Section 5.4).

**Datasets**. We conducted experiments on 4 simulation datasets and 1 real-world dataset. The simulation datasets are collected from the MultiGrid environment (Oguntola et al., 2023) and the Procgen benchmark (Cobbe et al., 2019). From Procgen, we use 3 environments: Bigfish, Leaper, and Starpilot. For a real-world dataset, we use nuPlan (Karnchanachari et al., 2024), an autonomous driving dataset. We use these datasets because they include multiple independent entities, such as agents, enemies, and cars. See details in Appendix A.

**Implementations**. As introduced in Sec. 4, the encoder is pretrained and frozen during latent action model learning. Rather than training a universal feature extractor across all datasets, we train a separate encoder for each dataset, as our work focuses on demonstrating the benefits of factorization for world models, rather than developing a universal world model. Dataset-specific encoder learning ensures that representation learning is not the performance bottleneck, enabling a meaningful comparison across different methods. Further FLAM implementation details are provided in Appendix B.

**Baselines**. We compare FLAM against the following baselines.

- **Genie** (Bruce et al., 2024) and **AdaWorld** (Gao et al., 2025) are vanilla latent action models that infer a single scene-level latent action. Genie constrains latent action capacity via vector quantization, while AdaWorld adopts a VAE-style regularization. Since Genie shares a similar implementation with LAPO, we omit LAPO from our comparisons.

- **World Model** uses ground-truth actions instead of inferred latent actions and only learns the forward dynamics model. It is evaluated only on simulation datasets where action labels are available.

- **PlaySlot** (Villar-Corrales & Behnke, 2025) first learns object-centric representations using SAVi (Kipf et al., 2021) and then trains a latent action model on top of the frozen representations.

- **SlotFormer** (Wu et al., 2022) similarly learns object-centric representations with SAVi, but trains a slot-based prediction model that conditions only on the current slots (i.e., without latent actions).

For a fair comparison, all methods use the same pretrained encoder for feature extraction and, when applicable, the same architectures for the factorizer, IDM, and FDM. In addition, to match latent action capacity across methods, while Genie and AdaWorld use a single latent action of dimension $d$, FLAM and PlaySlot use $K$ factor-wise latent actions of dimension $d/K$ each.

## 5.1. World Model Accuracy

We first evaluate FLAM's dynamics modeling accuracy, i.e., how accurately the model predicts future frames, and controllability, i.e., how much users can steer video generation via latent actions.

**Prediction Accuracy**. To assess dynamics modeling accuracy, we first infer latent actions $a_{1:T}$ from the *unseen* ground-truth frames $o_{1:T}$ and then generate $T$-step predictions $\hat{o}_{1:T}$ autoregressively. We measure prediction quality using peak signal-to-noise ratio (PSNR), learned perceptual image patch similarity (LPIPS), structural similarity index measure (SSIM), and Fréchet video distance (FVD). Finally, to contextualize the scores, we also report the reconstruction performance of the pretrained VQ-VAE on the future frames (denoted as **Recon**), which reflects the *best achievable quality* given the frozen encoder-decoder.

As shown in Tab. 1, FLAM achieves the best average performance and outperforms most baselines across datasets, highlighting the effectiveness of factored state representations and latent actions in multi-entity domains. Notably, FLAM even outperforms the World Model baseline. We attribute this result to two factors: in Procgen, only the player action is provided, so inferred latent actions help capture the stochastic motion of other entities (e.g., enemies); in MultiGrid, although actions for all agents are available, we follow the standard practice of treating them as a single monolithic action token. Compared to FLAM, World Model's underperformance further illustrate the limited generalization of monolithic models to state–action pairs that are rare or unseen during training. Meanwhile, as shown in Fig. 10-14, object-centric baselines such as PlaySlot perform competitively in the simple MultiGrid domain but degrade on datasets with more complex visuals, where they often fail to maintain temporally consistent slots. This inconsistency leads to unreliable latent action inference and inaccurate predictions. These results demonstrate the benefit of jointly learning representations and dynamics, rather than training them in separate stages. Appendix C.1 includes some example rollouts.

**Controllable video generation**. The factored latent actions learned by FLAM not only help with accurate world modeling, but can also serve as a manipulation surrogate to guide the video generation. We let human users specify an entity to control, together with sampling a latent action from the prior distribution for each time step, and then roll out multiple steps. Meanwhile, the remaining entities that are not manipulated would follow their original latent actions. As shown in Fig. 6, latent actions can be used as a control variable to generate various video frames even with the same initial frame.

*Table 1.* Prediction performance of all methods on simulation and real-world datasets. **Recon** reports the reconstruction quality of the pretrained VQ-VAE, serving as a reference upper bound for prediction performance.

| Dataset | Metric | FLAM (ours) | AdaWorld | Genie | World Model | PlaySlot | SlotFormer | Recon |
|---|---|---|---|---|---|---|---|---|
| Average | PSNR (↑) | **34.9** | 24.7 | 28.1 | N/A | 29.1 | 19.9 | 37.1 |
| | SSIM (↑) | **0.890** | 0.862 | 0.859 | N/A | 0.847 | 0.814 | 0.909 |
| | LPIPS (↓) | **0.051** | 0.096 | 0.091 | N/A | 0.120 | 0.223 | 0.035 |
| | FVD (↓) | **419.6** | 749.5 | 717.5 | N/A | 858.0 | 1597.7 | 215.1 |
| MultiGrid | PSNR (↑) | **56.5** | 24.0 | 33.9 | 56.1 | **56.5** | 24.9 | 56.5 |
| | SSIM (↑) | **0.944** | 0.909 | 0.933 | 0.943 | **0.944** | 0.912 | 0.944 |
| | LPIPS (↓) | **3e-4** | 0.08 | 0.014 | 6e-4 | **3e-4** | 0.056 | 3e-4 |
| | FVD (↓) | **2.9** | 913.6 | 177.5 | 5.5 | **2.9** | 451.6 | 2.5 |
| Bigfish | PSNR (↑) | **30.8** | 30.3 | 28.5 | 23.9 | 26.0 | 19.6 | 34.4 |
| | SSIM (↑) | **0.983** | 0.982 | 0.974 | 0.949 | 0.962 | 0.934 | 0.992 |
| | LPIPS (↓) | **0.011** | 0.012 | 0.021 | 0.049 | 0.038 | 0.221 | 0.004 |
| | FVD (↓) | 63.6 | **63.3** | 106.5 | 168.9 | 227.4 | 1455 | 16.8 |
| Leaper | PSNR (↑) | **38.1** | 23.3 | 35.0 | 24.7 | 27.5 | 21.4 | 40.3 |
| | SSIM (↑) | **0.976** | 0.924 | 0.973 | 0.924 | 0.945 | 0.913 | 0.977 |
| | LPIPS (↓) | **0.002** | 0.072 | 0.005 | 0.044 | 0.031 | 0.164 | 0.001 |
| | FVD (↓) | **12.1** | 348.1 | 27.3 | 127.3 | 361.5 | 1151 | 6.27 |
| Starpilot | PSNR (↑) | **29.3** | 27.8 | 27.0 | 25.8 | 18.3 | 19.8 | 32.6 |
| | SSIM (↑) | **0.971** | 0.965 | 0.962 | 0.959 | 0.886 | 0.907 | 0.980 |
| | LPIPS (↓) | **0.015** | 0.023 | 0.028 | 0.040 | 0.187 | 0.166 | 0.006 |
| | FVD (↓) | **73.4** | 113.4 | 141.0 | 194.9 | 792.0 | 720.9 | 24.7 |
| nuPlan | PSNR (↑) | **19.7** | 18.1 | 16.0 | N/A | 17.2 | 14.0 | 21.7 |
| | SSIM (↑) | **0.577** | 0.529 | 0.453 | N/A | 0.498 | 0.405 | 0.650 |
| | LPIPS (↓) | **0.229** | 0.292 | 0.389 | N/A | 0.344 | 0.508 | 0.163 |
| | FVD (↓) | **1946** | 2309 | 3135 | N/A | 2906 | 4210 | 1025 |

## 5.2. FLAM State Representation

Next, we evaluate whether FLAM factorizes state representations according to entities' actions. Intuitively, the learned factorization is jointly determined by the degree of action independence among entities and the number of factors $K$. When many entities have similar or correlated actions (e.g., a flock of birds flying together), a smaller $K$ encourages the model to merge them into the same factor, whereas a larger $K$ allows the model to allocate separate factors to individual entities and capture finer-grained action differences. Conversely, when entities act largely independently, FLAM requires a sufficiently large $K$ to model each entity accurately.

**Correlated Entities**. We construct a controlled MultiGrid dataset where two of four agents always execute identical actions at every timestamp (i.e., their action sequences are strongly correlated), while other aspects of the scene (e.g., positions) may vary. Using $K = 3$, we evaluate whether FLAM maps the two correlated agents to the same factor. As shown in Fig. 7, FLAM consistently assigns the correlated agents into a single factor, demonstrating that FLAM groups and separates entities based on their dynamics rather than visual appearance.

**Independent Entities**. On a separate MultiGrid dataset where four agents act independently, we evaluate whether

FLAM separates them into different factors with $K = 4$. We adopt the DCI (disentanglement, completeness, informativeness) metric (Eastwood & Williams, 2018) by treating each factor as a latent variable and measuring how well it encodes each agent's information with a linear probe (see details in Appendix C.2). High disentanglement and completeness indicates that each agent is primarily captured by a distinct factor, rather than being spread across multiple factors. High informativeness indicates that the corresponding factor captures a good amount of information about the agent and has good position prediction accuracy. As shown in Tab. 2, FLAM achieves the highest disentanglement and completeness, suggesting that it more reliably assigns each agent to an independent factors than the object-centric baselines. It also attains competitive informativeness, indicating that this improved factor–agent correspondence does not come at the cost of predictive accuracy. For reference, we also report the DCI scores obtained from randomly generated representations, denoted as **Random**.

**Number of factors** $K$. Finally, on the same domain with four independently acting agents, we study the effect of the number of factors by sweeping $K \in \{2, 4, 8, 16\}$.

As shown in Tab. 3, when $K$ is smaller than the number of entities, prediction performance degrades, as expected. Once $K$ is at least the number of entities, performance be-

*Table 2.* Factor-agent correspondence in the MultiGrid environments, evaluated by DCI (disentanglement, completeness, informativeness) metrics.

|            | FLAM(ours) | PlaySlot | SlotFormer | Random |
|------------|-----------|----------|-----------|--------|
| D (↑)      | **0.91**  | 0.81     | 0.85      | 0.32   |
| C (↑)      | **0.91**  | 0.81     | 0.85      | 0.30   |
| I (↑)      | **0.93**  | 0.86     | 0.90      | 0.10   |

*Table 3.* Prediction performance of FLAM on the MultiGrid dataset with 4 agents and different number of factors $K$.

| $K$        | 2     | 4        | 8        | 16       |
|------------|-------|----------|----------|----------|
| PSNR (↑)   | 53.4  | **56.5** | **56.5** | **56.5** |
| SSIM (↑)   | 0.943 | **0.944**| **0.944**| **0.944**|
| LPIPS (↓)  | 4e-4  | **3e-4** | **3e-4** | **3e-4** |
| FVD (↓)    | 3.62  | 2.87     | 2.80     | **2.73** |

comes stable across different choices of $K$, with negligible differences due to training stochasticity. Moreover, as shown in Fig. 8, even with excessive $K$, FLAM still consistently assigns each entity to a distinct factor across timestamps, demonstrating robustness to over-specifying $K$.

Therefore, for datasets where the number of entities varies across scenarios, one can choose a sufficiently large $K$ to cover the maximum number of entities, allowing unused factors to remain inactive when fewer entities are present.

### 5.3. Latent Action Policy Learning

We evaluate whether FLAM can facilitate policy learning from video with limited action supervision. First, for each environment, we train an expert policy using RL and collect an expert demonstration dataset containing 1M frames. Next, we train an action decoder on a small labeled subset of the demonstrations, where ground-truth actions are available, and we vary the labeled subset size (1k and 10k frames) to study label efficiency. We then use learned IDM and action decoder to infer pseudo action labels on the full 1M-frame demonstration dataset, yielding state-action pairs without using the true action labels for training. Finally, we train a behavior cloning policy on these pseudo-labeled data and evaluate its performance in unseen environments. To quantify the benefit of pseudo labels, we compare against a behavior cloning policy trained only on the labeled subset. For reference, we also report the performance of a random policy.

As shown in Tab. 4, FLAM provides the largest gains at the intermediate label size (10k). With very few action labels (1k), it is challenging to learn a reliable action decoder, so the resulting noisy pseudo labels provide limited benefit.

*Table 4.* Behavior cloning policy performance, evaluated by mean episodic return.

| Dataset          | FLAM    | vanilla BC | random |
|------------------|---------|-----------|--------|
| Bigfish (1k)     | **1.8** | 1.0       | 0.9    |
| Bigfish (10k)    | **8.8** | 3.6       |        |
| Starpilot (1k)   | 1.8     | **1.9**   | 1.1    |
| Starpilot (10k)  | **9.1** | 6.4       |        |

### 5.4. Ablation Studies

We further evaluate how architecture designs and hyperparameters affect FLAM's prediction performance.

**Factorizer architecture**. As described in Sec. 4.2, a key design of FLAM's factorizer is the causal temporal attention, which allows each slot to attend to its past values across timestamps and improves slot consistency. We ablate this component by removing temporal attention from the factorizer. Instead, the factorizer follows SAVi by computing slots autoregressively and initializing the slots at each timestamp using the slots from the previous timestamp. As detailed in Appendix. C.4, this removal significantly hurts prediction performance and slot consistency.

**Latent action model architecture**. In FLAM, for slot $i$, the IDM infers the latent action $a_t^i$ from $s_t^{1:K}$ and $s_{t+1}^i$, while the FDM predicts $s_{t+1}^i$ conditioned on $s_t^{1:K}$ and $a_t^i$. This design encourages per-slot action independence while still accounting for slot interactions. To evaluate this architectural choice, we compare against an ablation, where the IDM infers actions from all slots $(s_t^{1:K}, s_{t+1}^{1:K})$ and the FDM predicts $s_{t+1}^i$ conditioned on $(s_t^{1:K}, a_t^{1:K})$. As detailed in Appendix C.4, although conditioning on all slots achieves prediction performance similar to the original FLAM, it tends to assign multiple entities into the same factor since there is no explicit incentive to keep them separate, resulting in less disentangled representations.

**KL regularization coefficient** $\beta$ controls trade-off between latent action capacity and prediction quality. We report FLAM's prediction performance across different $\beta$ values. As detailed in Appendix C.4, FLAM is robust across a wide range of $\beta$ values, which makes it easy to tune.

## 6. Conclusion

We present FLAM, a factored latent action model that addresses a limitation of prior work in challenging multi-entity scenarios, where the underlying action space grows exponentially with the number of entities. Specifically, FLAM decomposes both the state and latent action representations into independent factors, and assumes all factor share in a common latent action space. Compared to prior work that must capture multi-entity behavior within a single monolithic latent action space, FLAM instead models each entity's action patterns within this shared space. This factor-

ized structure enables more accurate modeling of complex multi-entity dynamics and improves both prediction quality and controllability over prior methods.

**Limitation** In this work, we train a VQ-VAE from scratch for each dataset; adopting and fine-tuning pretrained tokenizers is a promising step toward sharing the encoder across datasets and moving toward a universal latent action model. Moreover, FLAM predicts in latent space using a transformer-based aggregator and relies on a pretrained decoder for visualization. Exploring more expressive decoders, such as diffusion or flow-matching models, could further improve the visual quality of generated rollouts.

## Impact Statement

This paper presents work whose goal is to advance the field of machine learning. There are many potential societal consequences of our work, none of which we feel must be specifically highlighted here.

**Acknowledgments** This work has taken place in the Learning Agents Research Group (LARG), Machine Intelligence through Decision-making and Interaction (MIDI) Lab, and Robot Interactive Intelligence (RobIn) Lab at the Artificial Intelligence Laboratory, The University of Texas at Austin. LARG research is supported in part by the National Science Foundation (FAIN-2019844, NRT-2125858), the Office of Naval Research (N00014-24-1-2550), Army Research Office (W911NF-17-2-0181, W911NF-23-2-0004, W911NF-25-1-0065), DARPA (Cooperative Agreement HR00112520004 on Ad Hoc Teamwork), Lockheed Martin, and Good Systems, a research grand challenge at the University of Texas at Austin. MIDI research is supported in part by the National Science Foundation (NSF-2340651, NSF-2402650), DARPA (HR00112490431), Army Research Office (W911NF-24-1-0193). The views and conclusions contained in this document are those of the authors alone. Peter Stone serves as the Chief Scientist of Sony AI and receives financial compensation for that role. The terms of this arrangement have been reviewed and approved by the University of Texas at Austin in accordance with its policy on objectivity in research.

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

## A. Dataset Details

We summarize here the datasets used in our experiments with additional implementation details. All datasets are split into train/validation/test with an 80-10-10 ratio by frame count, unless otherwise specified. Frames are normalized to $[0, 1]$ and resized to the image resolutions reported in Table 5.

**MultiGrid (Simulation).**   We adopt the MultiGrid environment (Oguntola et al., 2023) that supports multiple independently moving agents. Each video consists of a $8 \times 8$ grid world rendered at $128 \times 128$ resolution, with between 2 - 16 agents moving simultaneously. Unless specified otherwise, we use the dataset where all videos consist of 4 agents and the number of factors $K$ is set to 4.

**Procgen (Simulation).**   We adopt the Procgen benchmark (Cobbe et al., 2019) with background rendering disabled for consistency. Videos are rendered at $224 \times 224$ resolution.

**nuPlan (Real-world).**   We use the nuPlan benchmark (Caesar et al., 2022), restricting to front-facing camera streams only. Frames are resized to $224 \times 224$, and the number of factors is set to $K = 16$ to capture vehicles and pedestrians in each scene.

## B. Implementation Details

The hyperparameters used during encoder and factored latent action model training can be found in Table 5.

*Table 5.* Experiment hyperparameters. Simulation datasets are Multigrid and Procgen, while the real-world dataset is nuPlan.

| | Simulation | Real-world |
|---|---|---|
| Encoder architecture | IMPALA | MAGVIT-v2 |
| Decoder architecture | IMPALA | MAGVIT-v2 |
| Image size | 128 (Multigrid), 224 (Procgen) | 224 |
| $d_z$ | 128 | 128 |
| Tokenizer quantizer | FSQ | FSQ |
| Tokenizer codebook size | 1024 | 16384 |
| Tokenizer codebook levels | [4,4,4,4,4] | [4,4,4,4,4,4,4] |
| Optimizer | AdamW | AdamW |
| Learning rate | 1e-4 | 1e-4 |
| Batch size | 64 | 64 |
| Attention model dimension | 256 | 256 |
| Number of attention heads | 8 | 8 |
| Number of attention layers | 2 | 3 |
| Number of factors $K$ | 4 (Multigrid), 16 (Procgen) | 16 |
| KL regularization coefficient $\beta$ | 2e-4 | 2e-4 |
| Optimizer | AdamW | AdamW |
| Learning rate | 1e-4 | 1e-4 |
| Batch size | 32 | 32 |

**Computational Cost**   Let $N$ denote the number of patch-level features. FLAM's computation cost primarily consists of the attention between patches and slots, which scales as $O(NK)$, and interaction modeling among slots in the IDM/FDM, which scales as $O(K^2)$ per timestep. Since $K$ is chosen to reflect the number of high-level entities and is typically much smaller than the number of patch tokens $N$, this slot-level interaction cost is modest in our experiments. In contrast, dense patch-level video models often perform attention over all patch tokens, which scales as $O(N^2)$.

# C. Experiments

## C.1. World Model Accuracy

We evaluate FLAM from two perspectives, dynamics modeling accuracy, i.e., how accurately the model predicts future frames, and controllability, i.e., how much users can steer video generation via latent actions.

**Prediction Accuracy**    To assess dynamics modeling accuracy, we first infer latent actions $a_{1:T}$ from the ground-truth frames $o_{1:T}$ and then generate $T$-step predictions $\hat{o}_{1:T}$ autoregressively, where we use $T = 10$ across all datasets. In addition to the quantitative results in Sec. 5.1, we visualize example rollouts in Fig. 10-14. Across most datasets, FLAM generates more consistent factor-entity bindings and more accurate predictions than the baselines.

**Scaling with the number of entities**    To investigate whether FLAM facilitates dynamics modeling in complex multi-entity videos, we conduct an ablation study on MultiGrid by varying the number of entities. For a fair comparison, we match the total latent action capacity between FLAM and AdaWorld. For instance, on the 8-agent setting, AdaWorld uses a single 256-dimensional latent action, while FLAM uses eight 32-dimensional latent actions. For Genie, we similarly ensure its latent action capacity by matching the codebook size to the number of joint action combinations. As shown in Fig. 4, as the number of entities increases, FLAM remains substantially more robust than a non-factored LAM.

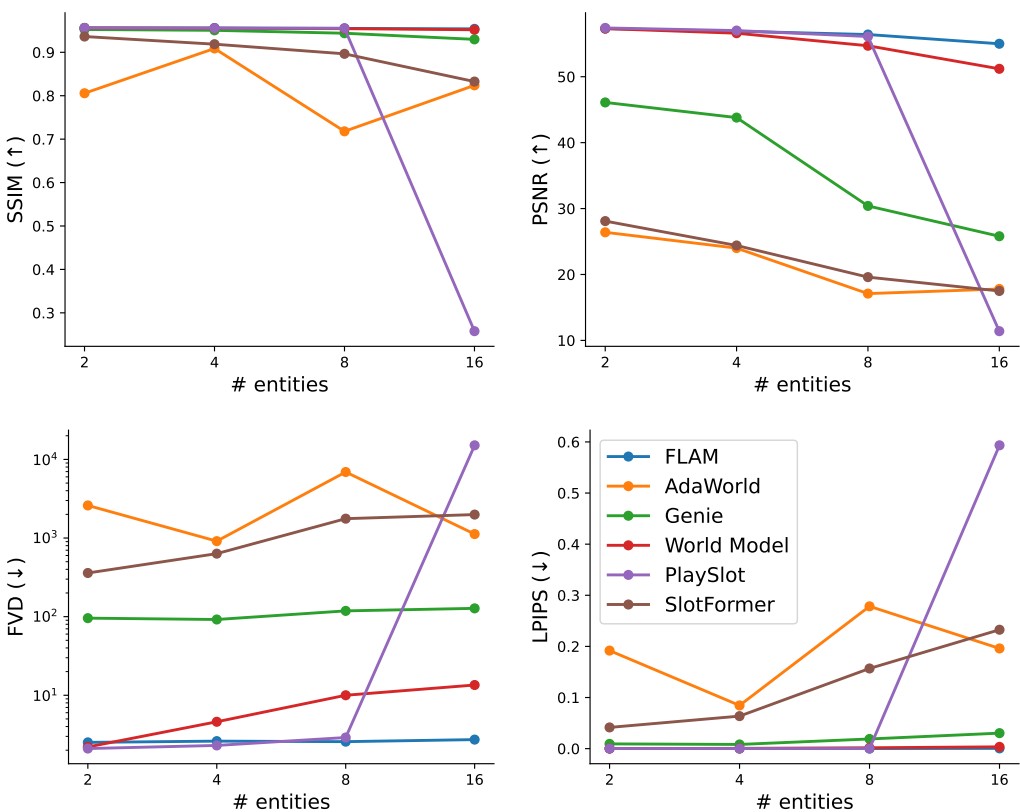

*Figure 4.* Prediction performance variation along with increasing number of entities in the scene.

**Controllable video generation**    The factored latent actions learned by FLAM not only help with accurate world modeling, but can also serve as a manipulation surrogate to guide the video generation. We let human users specify an entity to control, together with sampling a latent action from the prior distribution for each time step, and then roll out multiple steps. Meanwhile, the remaining entities that are not manipulated would follow their original latent actions. An example of generated videos is shown in Fig. 6. The first row is the original video. Take the first frame of the original video as the initial start, each of the following rows represents the generated video of manipulating one entity. It demonstrates that latent actions can be used as a control variable to generate various video frames even with the same initial frame. This indicates

that factorization offers the freedom of manipulating each entity independently instead of the monolithic scene, therefore leading to more diversity in video generation.

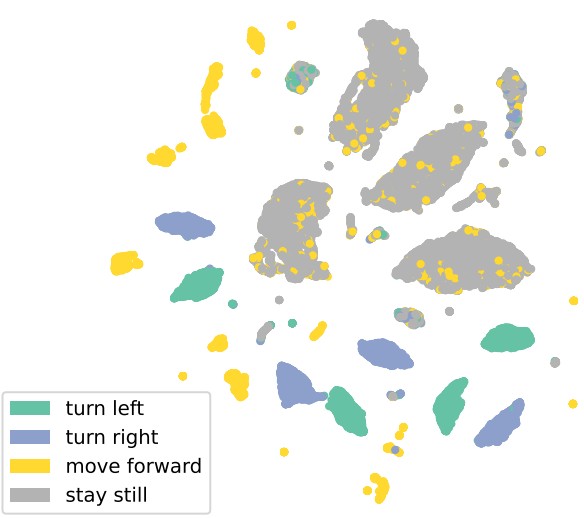

*Figure 5.* UMAP projection of the learned latent actions on the MultiGrid dataset. Each point corresponds to a latent action inferred by the IDM for one factor on a single transition from the observation-only dataset. Points are colored by the ground-truth actions taken by the corresponding agent at that transition (action labels are used only for visualization and are not used for training).

To demonstrate that FLAM learns latent actions that are well separated across different ground-truth actions and thus easy for humans to specify, we visualize the extracted latent actions for all agents by projecting them to 2D with UMAP (McInnes et al., 2018). As shown in Fig. 5, the latent actions form well-separated clusters that align with the true actions, with only a small number of `move forward` samples overlapping with `stay still` clusters. This overlap is expected, as, in many scenarios, an agent may take a `move forward` action but remain stationary due to being blocked. In such cases, the observed transition is indistinguishable from `stay still`, and the inferred latent action accordingly falls into the `stay still` cluster.

### C.2. FLAM State Representation

**Correlated Entities**   We construct a controlled MultiGrid dataset where two of four agents always execute identical actions at every timestamp (i.e., their action sequences are strongly correlated), while other aspects of the scene (e.g., positions) may vary. Using $K = 3$, we evaluate whether FLAM maps the two correlated agents to the same factor. As shown in Fig. 7, FLAM is able to group the entities that share the same action into the same factor, while separating the entities with different actions, based on their dynamics rather than visual appearance.

**Independent Entities**   In this section, we provide details of the DCI (disentanglement, completeness, informativeness) evaluation used in Sec. 5.2.

We use an unseen MultiGrid dataset with 4 agents for evaluation, and all evaluated methods use $K = 4$ factors during training. To compute the DCI score, we first extract slots $s^{1:K} \in \mathbb{R}^{K \times d}$ for each frame. We also record the agent position $y^{1:K}$, each $y^i \in \{1, \ldots, C\}$ is a discrete scalar indicating which grid the agent $i$ is at among $C$ potential locations. Then we split the data into a train set for probe learning and a validation sets for DCI computation.

During training, we learn a linear classifier probe $f : \mathbb{R}^d \to \Delta^C$ that predicts the agent position from each slot. Slots are permutation-equivariant, i.e., slot $i$ can correspond to different agents across frames. We therefore learn probes using permutation-invariant assignment. To compare a slot $i$ against a candidate agent $j$, we use the negative log-likelihood as the

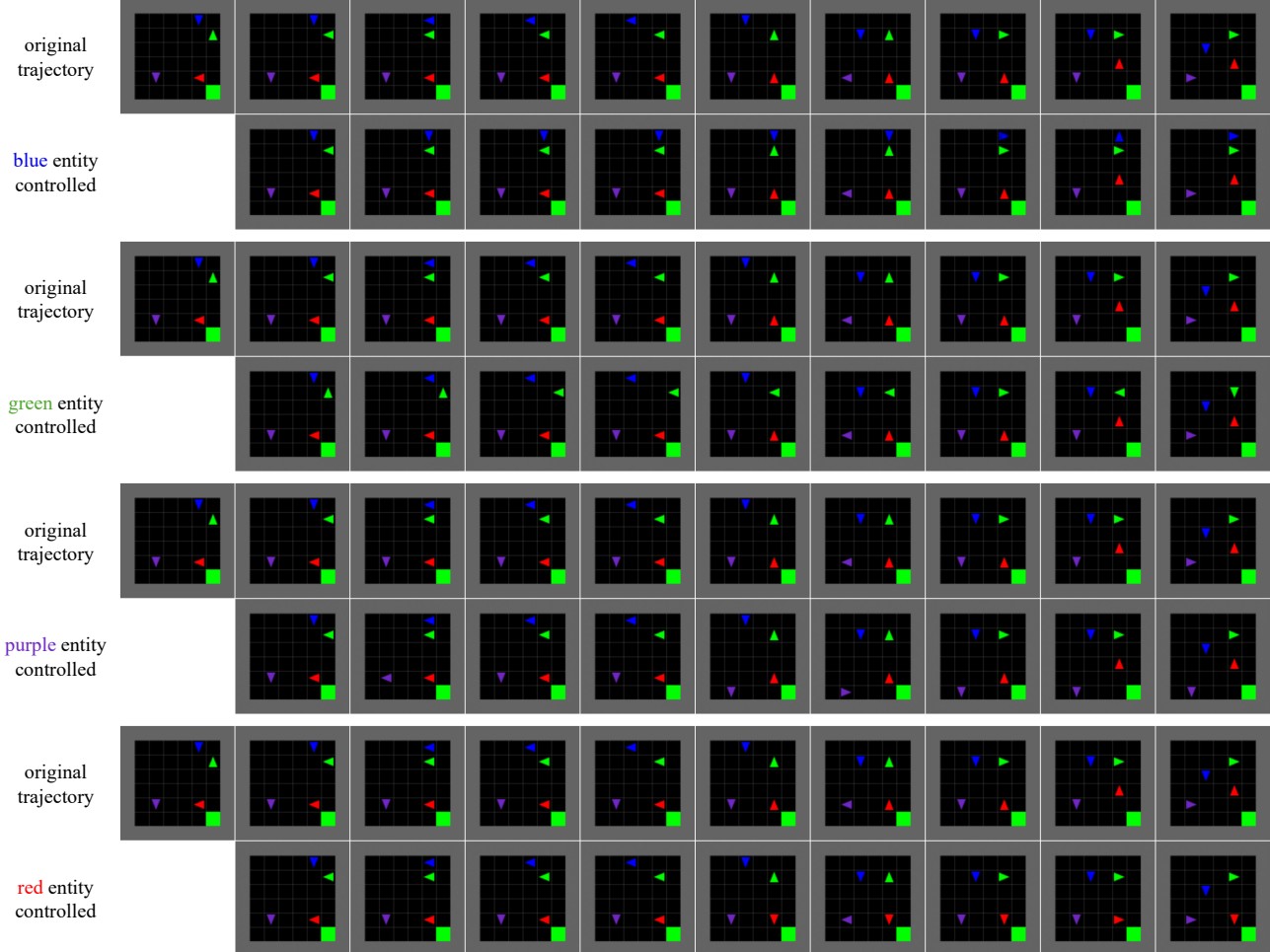

*Figure 6.* FLAM allows for changing / editing the motion of one entity without affecting the others. Here we show controllable video generation on MultiGrid through specifying latent actions for one of the movable agents, while other agents follow their original inferred latent actions.

cost and solve a matching problem to enforce a one-to-one correspondence $\tau$ between slots and agents:

$$\tau = \arg\min_{\tau \in S_K} \sum_{i=1}^{K} -\log f(y^{\tau(i)}|s^i),$$

where $S_K$ is the set of permutations of $\{1, \ldots, K\}$ and $\tau$ is computed with the Hungarian algorithm. Given $\tau$, the probe is trained with cross-entropy on the matched labels.

After training, we follow (Eastwood & Williams, 2018) to compute DCI metrics as follows.

- **Disentanglement and completeness**: To quantify whether each slot corresponds to a single agent (disentanglement) and each agent is covered by a single slot (completeness), we convert likelihood into soft assignment distributions. We define disentanglement and completeness distributions:

$$P_{\text{dis}}^{ij} = \frac{f(y^j|s^i)}{\sum_{k=1}^{K} f(y^k|s^i)},$$

$$P_{\text{comp}}^{ij} = \frac{f(y^j|s^i)}{\sum_{k=1}^{K} f(y^j|s^k)}.$$

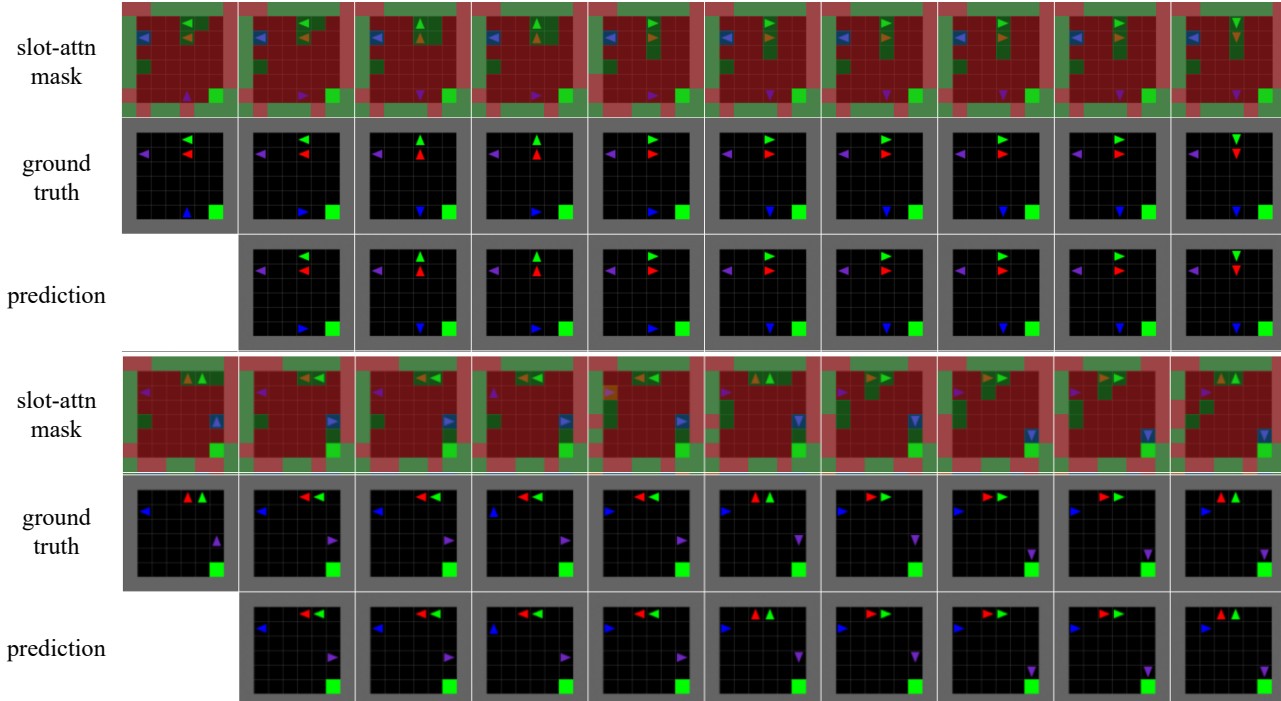

*Figure 7.* Two example rollouts where two entities (green and red ones) share the same actions. FLAM consistently maps the two correlated agents to the same factor.

We then compute disentanglement and completeness using normalized entropy with base $K$:

$$D = 1 + \sum_{j=1}^{K} P_{\text{dis}}^{ij} \log_K P_{\text{dis}}^{ij},$$

$$C = 1 + \sum_{i=1}^{K} P_{\text{comp}}^{ij} \log_K P_{\text{comp}}^{ij}.$$

We report $D$ and $C$ averaged over all slots / agents and over all frames in the validation set.

- **Informativeness**: We measure informativeness as the classification accuracy under the optimal matching and report $I$ averaged over all frames in the validation set.

$$I = \frac{1}{K} \sum_{i=1}^{K} \mathbb{1}\left[\arg\max_c p_i(c) = y^{\tau(i)}\right]. \tag{8}$$

**Number of factors $K$**   Finally, on the same domain with four independently acting agents, we study the effect of the number of factors by sweeping $K \in \{2, 4, 8, 16\}$.

In datasets where the number of entities varies across scenarios, one can choose a sufficiently large $K$ to cover the maximum number of entities, allowing unused factors to remain inactive when fewer entities are present.

### C.3. Latent Action Policy Learning

During data collection, for each Procgen environment, we train an expert policy using Phasic Policy Gradient (Cobbe et al., 2021) on all levels for 25M environment steps. We then collect demonstrations on disjoint sets of levels for training, validation, and testing, ensuring that the behavior cloning policy is evaluated on unseen environments. For a fair comparison, FLAM and the vanilla behavior cloning baseline use the same policy architecture and training setup, and the only difference is that FLAM additionally trains on pseudo action labels inferred from unlabeled videos.

*Table 6.* Prediction performance of FLAM and its ablative variants on the MultiGrid dataset.

| Metric | FLAM (ours) | FLAM w/o Factorizer Temporal Attention | Global-Coupled FLAM |
|---|---|---|---|
| PSNR (↑) | **56.5** | 51.6 | 56.3 |
| SSIM (↑) | **0.944** | 0.941 | **0.944** |
| LPIPS (↓) | **3e-4** | 3e-3 | 4e-4 |
| FVD (↓) | **2.9** | 36.6 | 3.72 |

*Table 7.* Factor-agent correspondence of FLAM and its ablative variants on the MultiGrid dataset, evaluated by DCI (disentanglement, completeness, informativeness) metrics.

| | FLAM(ours) | FLAM w/o Factorizer Temporal Attention | Global-Coupled FLAM | Random |
|---|---|---|---|---|
| D (↑) | **0.91** | 0.33 | 0.79 | 0.32 |
| C (↑) | 0.91 | 0.23 | **1.00** | 0.30 |
| I (↑) | **0.93** | 0.20 | 0.58 | 0.10 |

## C.4. Ablation Studies

We further evaluate how architecture designs and hyperparameters affect FLAM's prediction performance.

**Factorizer architecture**    As described in Sec. 4.2, a key design of FLAM's factorizer is the causal temporal attention, which allows each slot to attend to its past values across timestamps and improves slot consistency. We ablate this component by removing temporal attention from the factorizer. Instead, the factorizer follows SAVi by computing slots autoregressively and initializing the slots at each timestamp using the slots from the previous timestamp. Then we evaluate its impact on prediction performance and representation quality.

As shown in Tab. 6 and Fig. 9, removing temporal attention leads to temporally inconsistent factor assignments and degraded predictions. Moreover, Tab. 7 shows that this ablation substantially reduces factor-entity correspondence, demonstrating the importance of temporal attention in maintaining consistent bindings over time.

**Latent action model architecture**    In FLAM, for slot $i$, the IDM infers the latent action $a_t^i$ from $s_t^{1:K}$ and $s_{t+1}^i$, while the FDM predicts $s_{t+1}^i$ conditioned on $s_t^{1:K}$ and $a_t^i$. This design encourages per-slot action independence while still accounting for slot interactions. To evaluate this architectural choice, we compare against an ablation, **Global-Coupled FLAM**, where the IDM infers actions from all slots $(s_t^{1:K}, s_{t+1}^{1:K})$ and the FDM predicts $s_{t+1}^i$ conditioned on $(s_t^{1:K}, a_t^{1:K})$.

As shown in Tab. 6, this ablation achieves prediction performance similar to the original FLAM, as it retains the same overall capacity for encoding transition information. However, because both the IDM and FDM condition on all slots and all latent actions, the model has little incentive to maintain a one-to-one correspondence between entities and factors. In particular, the FDM can attend to $a_t^j$ when predicting $s_{t+1}^i$ and still achieve accurate prediction. Consequently, as shown in Tab. 7 and Fig. 9, Global-Coupled FLAM tends to merge multiple entities into the same factor, yielding less disentangled representations.

**KL regularization coefficient** $\beta$    controls the trade-off between latent action capacity and prediction quality. We report prediction performance across different $\beta$ values to quantify FLAM's sensitivity to this hyperparameter. As shown in Tab. 8, FLAM is robust across a wide range of $\beta$ values, which makes it easy to tune.

*Table 8.* Prediction performance of FLAM on the MultiGrid dataset with different KL regularization coefficient $\beta$ values.

| Metric | $\beta = 1e-5$ | $\beta = 1e-4$ | $\beta = 1e-3$ | $\beta = 1e-2$ |
|---|---|---|---|---|
| PSNR (↑) | **56.5** | **56.5** | 56.3 | 24.9 |
| SSIM (↑) | **0.944** | **0.944** | **0.944** | 0.911 |
| LPIPS (↓) | **3e-4** | **3e-4** | 4e-4 | 0.057 |
| FVD (↓) | **2.84** | 3.40 | 3.78 | 479.9 |

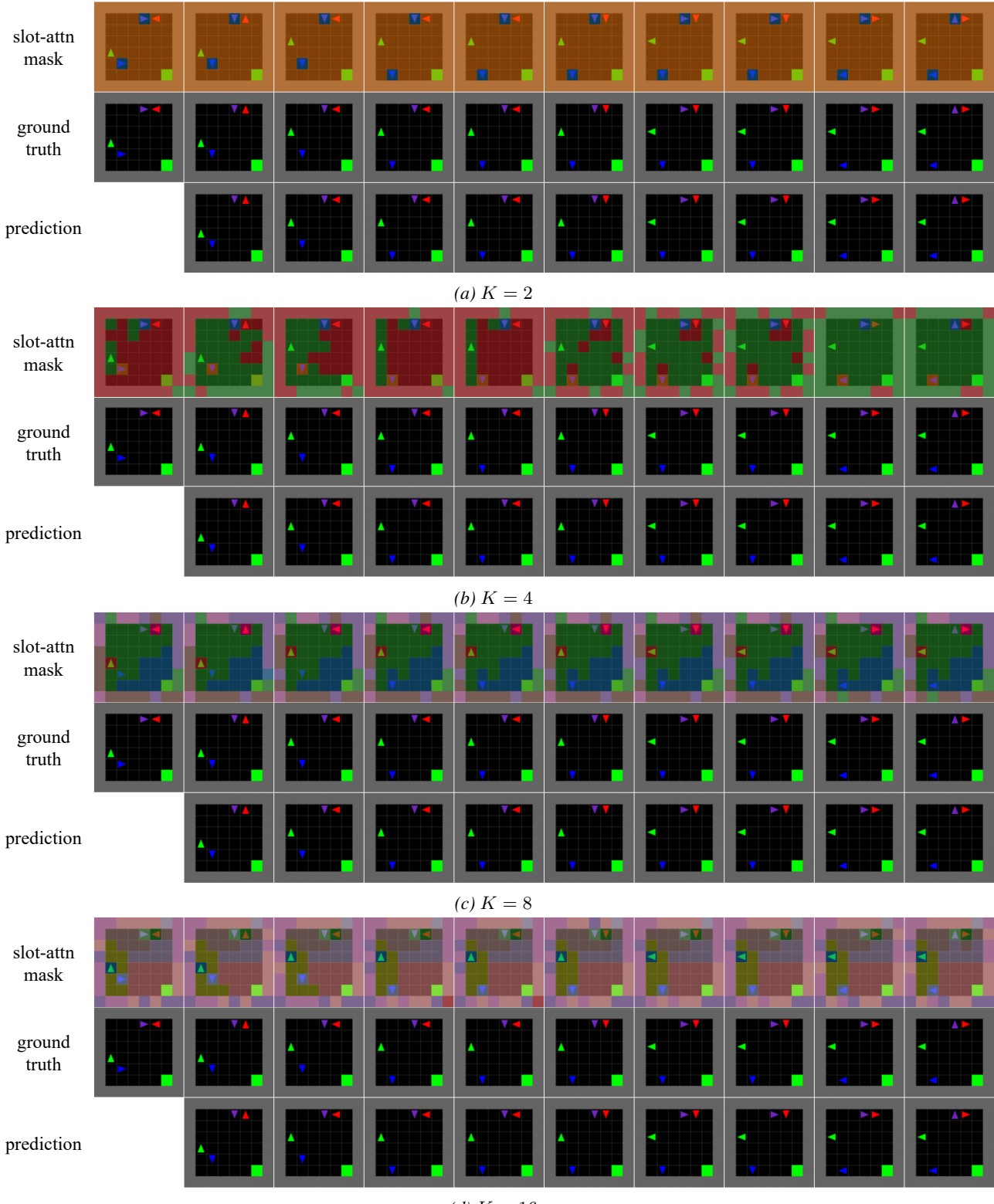

*Figure 8.* Ablation of FLAM on 4-agent MultiGrid dataset, with $K \in \{2, 4, 8, 16\}$. When $K$ is less than the number agents, FLAM assigns two agents to each factor to maximize latent action utilization. When $K$ is equal to or greater than the number of agents, FLAM consistently assigns each entity to a distinct factor, even with excessive $K$, demonstrating FLAM's robustness to over-specifying $K$.

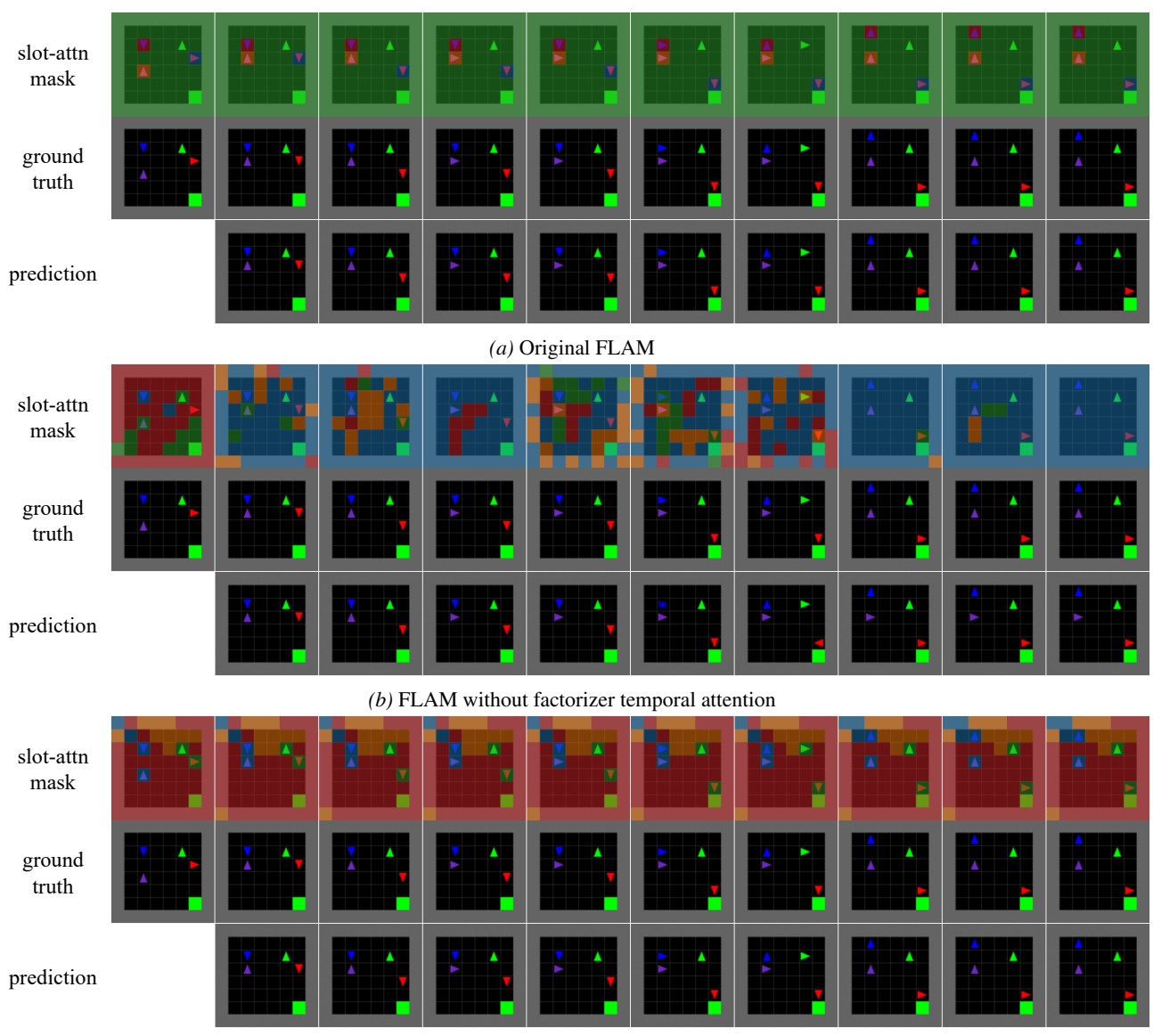

*(a)* Original FLAM

*(b)* FLAM without factorizer temporal attention

*(c)* Global-Coupled FLAM

*Figure 9.* Architecture ablations of FLAM on the 4-agent MultiGrid dataset. Removing temporal attention from the factorizer leads to temporally inconsistent factor assignments and degraded long-horizon predictions (notably in the last few timestamps). In contrast, the Global-Coupled variant can produce accurate predictions but tends to assign multiple entities into the same factor, resulting in poorer disentanglement than the original FLAM implementation.

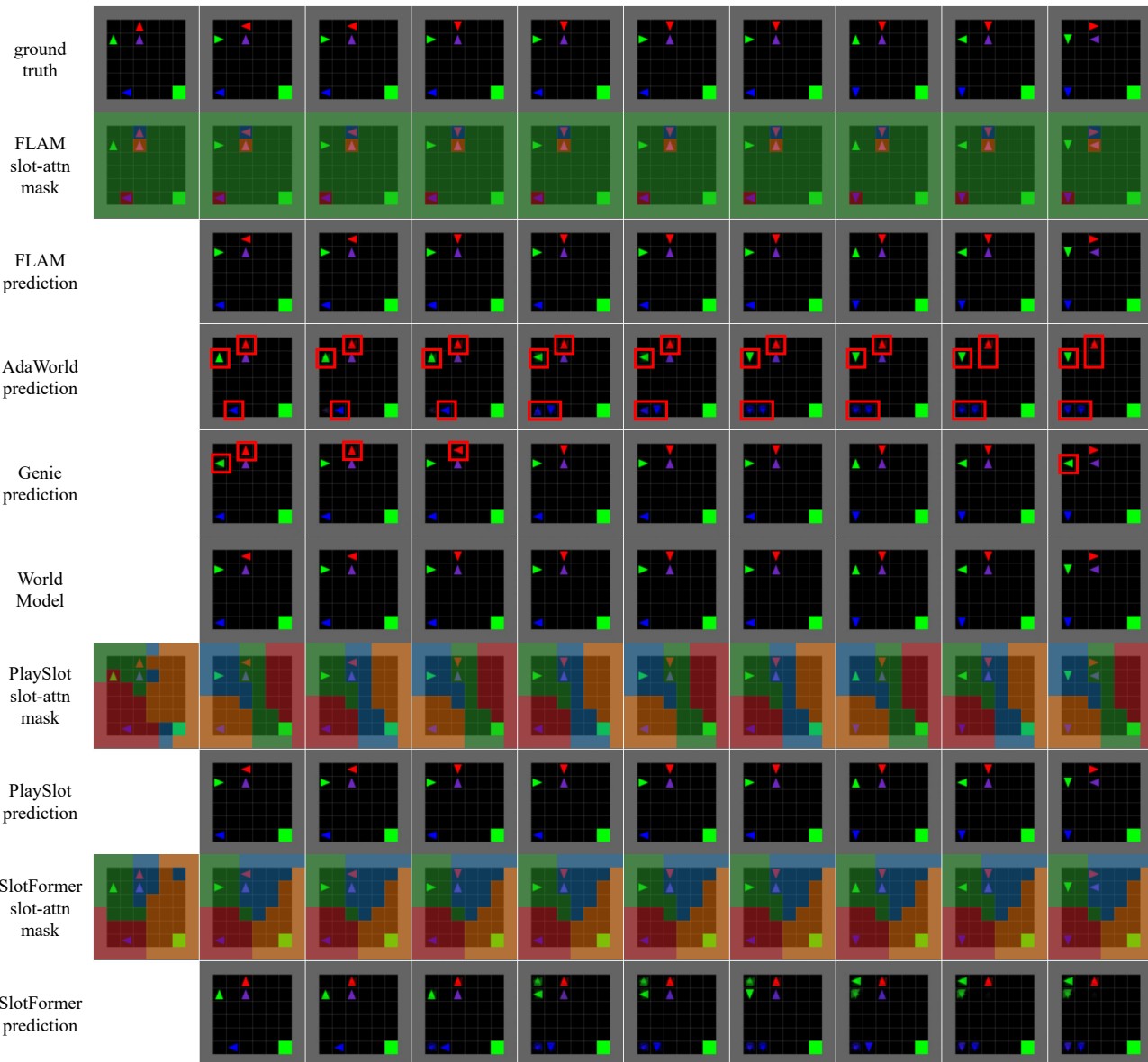

*Figure 10.* Prediction rollouts of all methods on the MultiGrid dataset, with inaccuracies highlighted by red boxes (except for SlotFormer, where the errors are visually apparent). When applicable, we also show the slot-patch binding masks from slot attention, displayed on top of the ground-truth observations. FLAM maintains consistent factor-entity bindings over time and generates accurate predictions. In contrast, AdaWorld and SlotFormer fail to generate accurate predictions from the start, and Genie makes mistakes at a few steps. Meanwhile, PlaySlot and SlotFormer rely on object-centric representations learned from reconstruction which incorrectly bind the red and purple entities to the same slot.

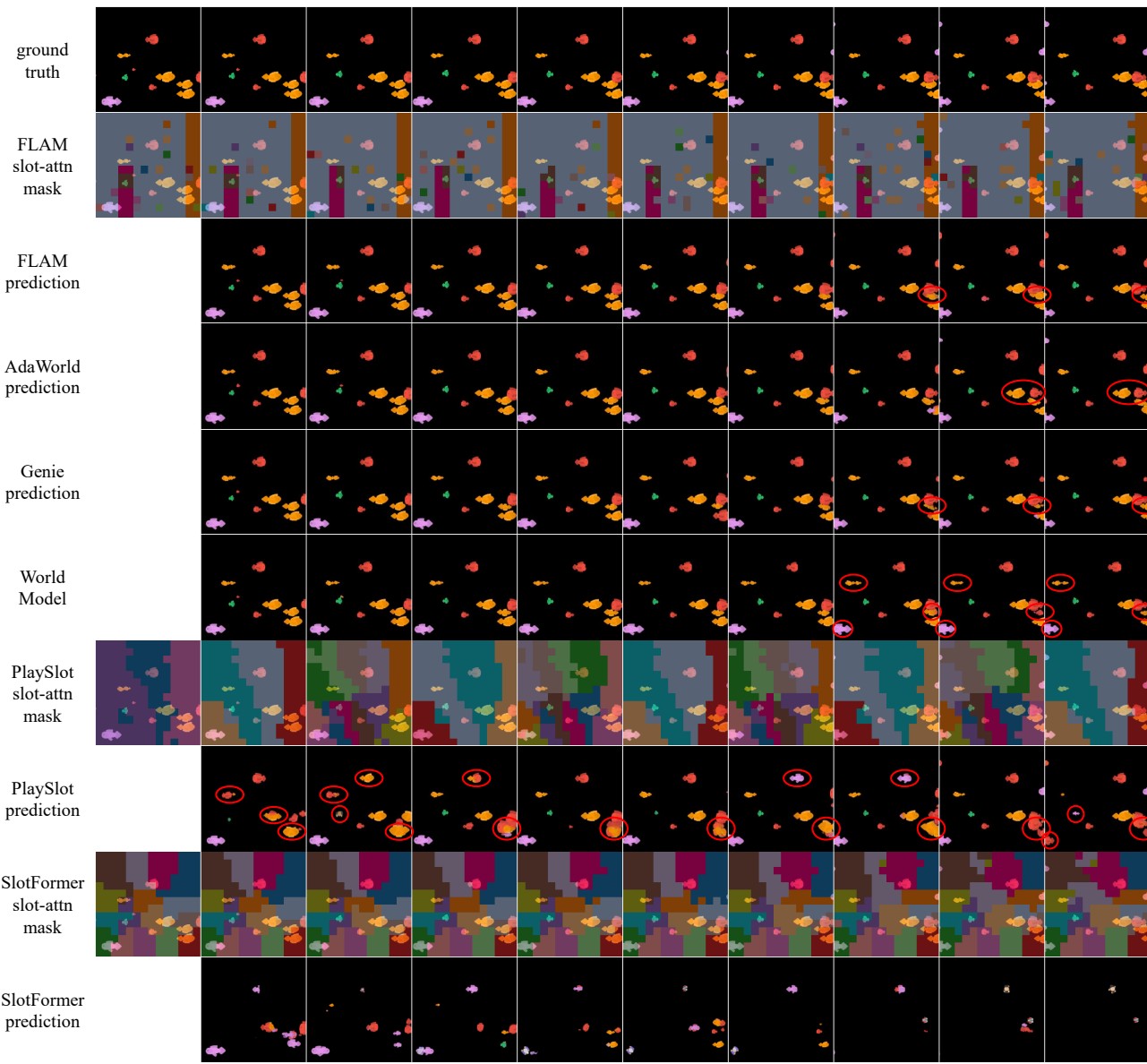

*Figure 11.* Prediction rollouts of all methods on the Bigfish dataset, with inaccuracies highlighted by red circles (except for SlotFormer, where the errors are visually apparent). When applicable, we also show the slot-patch binding masks from slot attention, displayed on top of the ground-truth observations. FLAM maintains consistent factor-entity bindings over time, e.g., the brown slot consistently attends to the green fish controlled by the player. In terms of prediction quality, FLAM and Genie only make errors on the occluded fish. AdaWorld to track the speeds of the circled yellow and red fish, which move more slowly than in the ground truth. The World Model baseline also struggles to capture the motion of non-controllable fish, since only the player's action is available. PlaySlot and SlotFormer rely on reconstruction-based object-centric representations that do not yield consistent slot assignments. As a result, PlaySlot fails to preserve consistent shapes and colors for multiple fish.

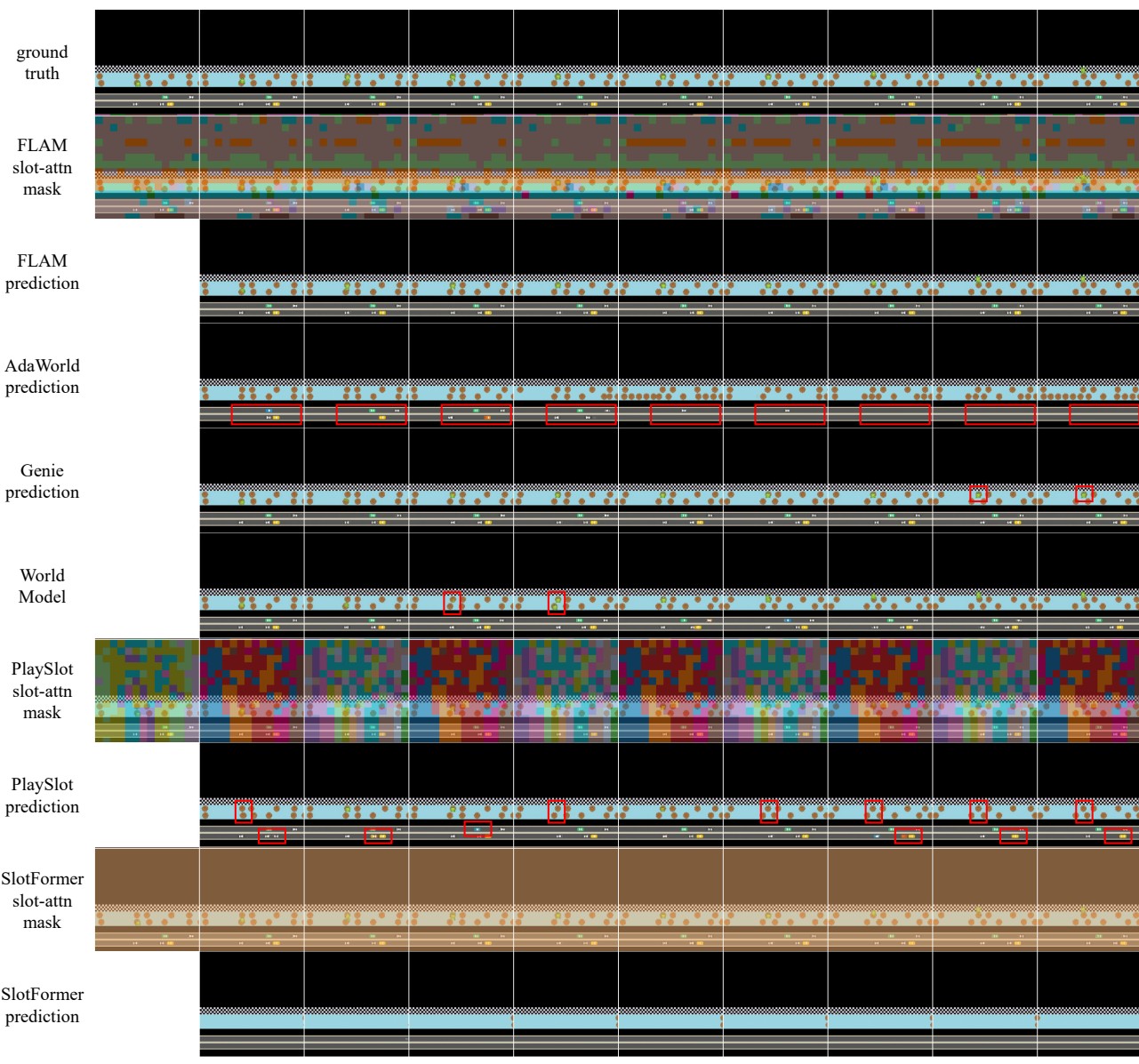

*Figure 12.* Prediction rollouts of all methods on the Leaper dataset, with inaccuracies highlighted by red boxes (except for PlaySlot and SlotFormer, where the errors are visually apparent). When applicable, we also show the slot-patch binding masks from slot attention, displayed on top of the ground-truth observations. FLAM generates accurate predictions, in particular capturing how the frog jumps between logs. Genie and World Model achieve the second best prediction quality, but fails to model the frog's jumps in some timestamps. In contrast, AdaWorld and PlaySlot generate inaccurate predictions from the start, either removing the frog and cars from the scene or hallucinating non-existent.

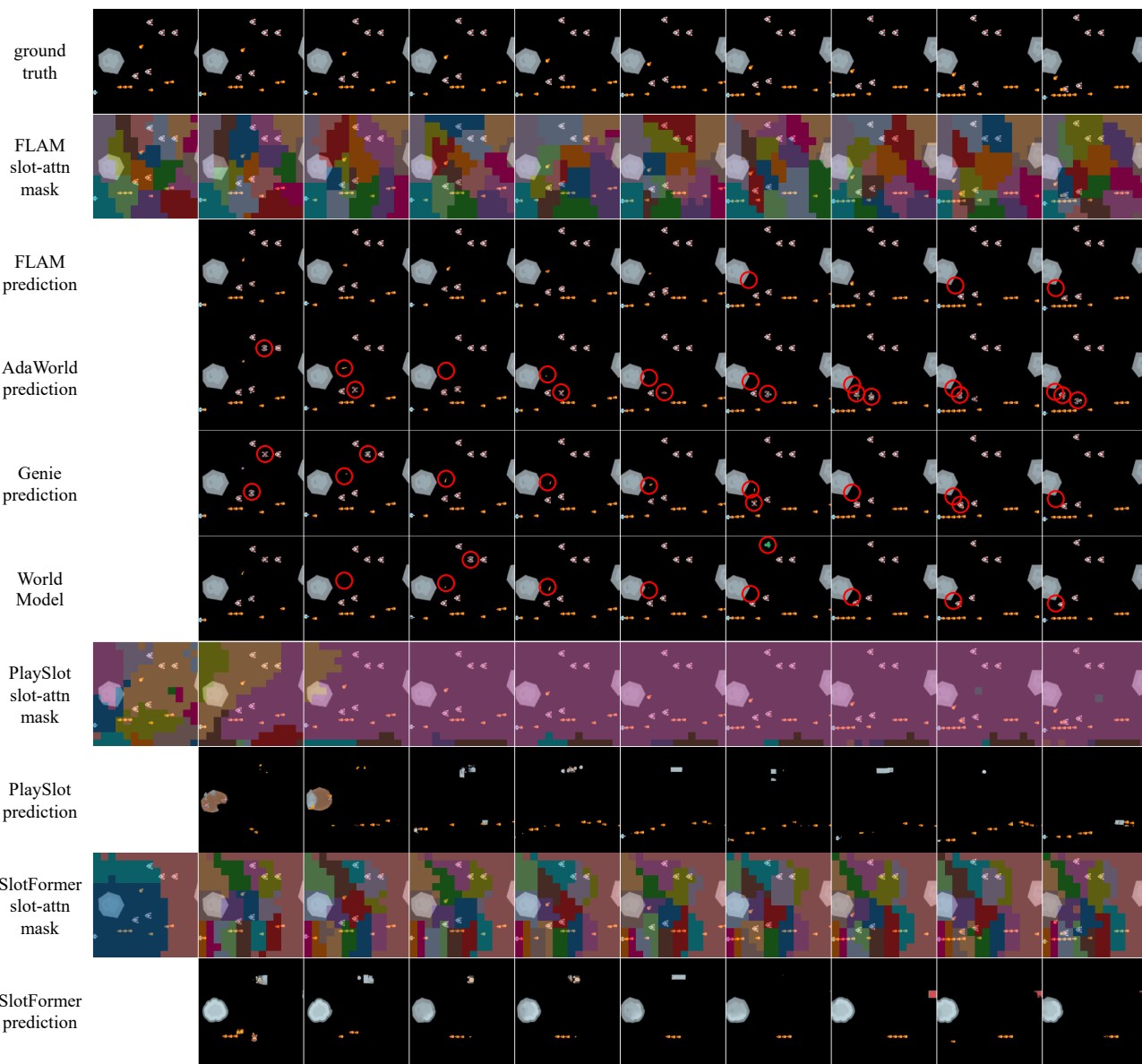

*Figure 13.* Prediction rollouts of all methods on the Starpilot dataset, with inaccuracies highlighted by red circles (except for PlaySlot and SlotFormer, where the errors are visually apparent). When applicable, we also show the slot-patch binding masks from slot attention, displayed on top of the ground-truth observations. FLAM maintains a consistent binding for the player in the bottom-left corner (shown in cyan) and generates accurate predictions, except for the enemy bullet heading toward the player. Other entities (e.g., bullets, enemies, and meteors) move at constant speed, so even with less consistent bindings, FLAM can still achieve reasonable predictions. In contrast, AdaWorld, Genie, and the World Model baseline generate inaccurate predictions from the start, frequently shifting enemy appearances and missing the enemy bullet toward the player.

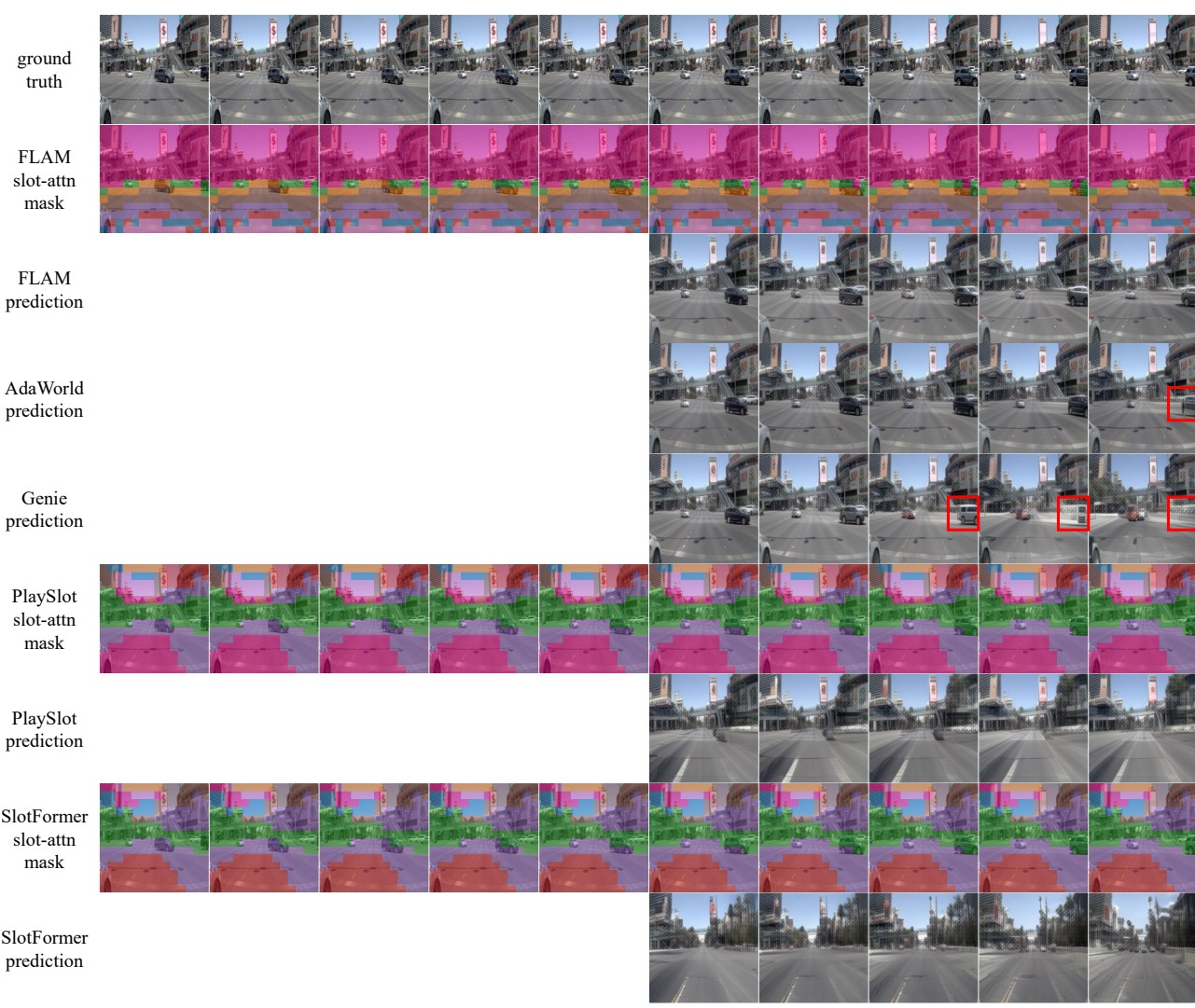

*Figure 14.* Prediction rollouts of all methods on the nuPlan dataset, with inaccuracies highlighted by red boxes (except for SlotFormer, where the errors are visually apparent). Models are conditioned on a fixed history window of size $w = 5$. When applicable, we also show the slot-patch binding masks from slot attention, displayed on top of the ground-truth observations. Compared to the baselines, FLAM generates relatively accurate predictions for the grey car moving from the center to the right. In contrast, AdaWorld and Genie fail to maintain a consistent appearance for this moving car. Meanwhile, PlaySlot and SlotFormer fail to model the dynamics and largely ignore the cars in the scene.

