# OpenReview forum: "Factored Latent Action World Models"
_ICML.cc/2026/Conference — ICML 2026 regular_

### Official Review · Reviewer_3Lbk · 2026-03-11

**Soundness:** 2
**Presentation:** 3
**Significance:** 2
**Originality:** 2
**Overall Recommendation:** 4
**Confidence:** 3

**Summary:**

This work extends the adaptability of LAM to scenarios involving multiple motion entities by decomposing the scene into independent factors, allowing each factor to independently infer its latent action and predict the next-step factor value. The method is evaluated on both simulated and real-world multi-entity datasets, and the results demonstrate the superiority of FLAM.

**Compliance With Llm Reviewing Policy:**

Affirmed.

**Final Justification:**

The detailed rebuttal and additional experiments resolved most of my concerns.

**Key Questions For Authors:**

See Strength and weakness

**Limitations:**

See Strength and weakness

**Strengths And Weaknesses:**

Strength
1, FLAM unifies the modeling of latent actions and state representations, enabling each entity to possess an independent action representation.  This design introduces a stronger structural inductive bias, improving the accuracy of LAM when modeling more complex multi-entity scenarios

2, The experimental evaluation is relatively comprehensive, validating the performance of FLAM across multiple simulated and real datasets using several prediction metrics such as PSNR and SSIM.

Weakness
1, However, several components used in this work, such as slot attention and VQ-VAE, are existing techniques, and FLAM mainly combines these prior methods, resulting in limited core methodological novelty.Moreover, the construction of FLAM relies on the assumption that the actions of different entities can be well decomposed into independent factors.  In many real-world scenarios, however, entities often exhibit strong interaction and coupling, such as in crowd behavior or collaborative robotics. It remains unclear whether the proposed method would still be effective in such scenarios.

2, In real-world settings, the experiments reported in the paper mainly focus on video prediction metrics. They do not clearly demonstrate the impact of FLAM on planning, control, or behavior prediction performance. For instance, in a real traffic scenario, one could fix the current intersection scene and only modify the latent actions of vehicles on the left or right to observe whether the model generates reasonable behavior while keeping the actions of other vehicles unchanged. Such an experiment would provide more convincing evidence.

3, As the number of entities in a video increases, the number of slots also grows accordingly. The paper lacks analysis comparing FLAM with existing methods in terms of GPU memory consumption, inference time, and FLOPs. It also does not discuss how the number of slots scales with memory requirements.

Q:
1. In real-world environments, the color of the subject in video prediction undergoes noticeable changes. What factors might cause this, such as the car's rearview mirror? Is it due to insufficient modeling of spatial understanding or confusion caused by the modeling of object opposition and decoupling?
2. It seems that the method's greatest benefit comes from Factorizer Temporal Attention. How is this factor defined, and does a factor of different complexity levels need to be used for videos of different complexities?

---

> ### Author Rebuttal · Authors · 2026-03-31
>
> We appreciate the reviewer’s time and constructive feedback.
>
> > `W1a` FLAM uses existing components such as Slot Attention and VQ-VAE, so the core novelty appears limited.
>
> `A1` We thank the reviewer for this thoughtful comment. Our novelty claim is not that these building blocks are individually new, but that we introduce a factorized latent action formulation for world modeling from action-free video that addresses a key limitation of prior latent action models in multi-entity scenes. Thus, the novelty lies in the formulation, training paradigm, and inductive bias of the overall method, rather than in inventing new low-level modules.
>
> > `W1b` FLAM assumes factorized actions, which may be problematic in strongly coupled real-world scenarios.
>
> `A2` We thank the reviewer for raising this important point. FLAM does not require every entity to always be assigned to a separate factor. Instead, it learns an action-driven factorization: when entities act independently, separating them helps, while when their dynamics are strongly correlated, FLAM can group them into the same factor if that better supports prediction. This is already partially shown in Section 5.2, where two coordinated agents that always share the same action are consistently assigned to one factor.
>
> To further validate this point, we ran additional experiments on Hallway, a MultiGrid environment where agents must collaborate to pick up keys and open doors.
>
> | Metric | FLAM | AdaWorld |
> |---|---:|---:|
> | PSNR (↑) | 31.5 | 16.5 |
> | SSIM (↑) | 0.956 | 0.708 |
> | LPIPS (↓) | 0.019 | 0.203 |
> | FVD (↓) | 50.5 | 135.0 |
>
> FLAM outperforms AdaWorld. Due to limited computation, we will add other baselines in the next version. This suggests that factorized latent actions remain effective beyond mostly independent scenes. More broadly, FLAM benefits from factorization when interactions are localized, while still being able to group tightly coupled entities when needed.
>
> > `W2` The real-world experiments focus mainly on prediction metrics and do not clearly demonstrate impact on control or behavior.
>
> `A3` We thank the reviewer for this helpful suggestion. Following it, we ran an additional intervention-based experiment on a real nuPlan scene: https://imgur.com/a/jHAL4WW. The first row shows the ground-truth future frames, the second row shows the rollout with the original inferred latent actions, and the third row shows the rollout after randomly resampling the latent action of a single factor while keeping all others fixed.
>
> We observe a localized change: it mainly slows the rightward motion of the black car, while leaving the white car largely unaffected. This provides direct evidence that FLAM supports selective control in a real-world traffic setting, beyond the controllability results already shown in MultiGrid. We will include this experiment in the revised paper.
>
> > `W3` The paper lacks analysis of GPU memory, inference time, FLOPs, and slot-scaling cost.
>
> `A4` We thank the reviewer for this helpful suggestion and will add discussion in the revision. Analytically, the dominant interaction cost in FLAM comes from attention over slots, so the cost is $O(K^2)$. This should be viewed relative to patch-level video models, where attention typically scales with the number of patches, often much larger than $K$. Thus, factorization can potentially reduce computation by replacing dense patch-level interaction with a smaller number of entity-level factors.
>
> > `Q1` In real-world environments, predicted subject colors sometimes change noticeably. What causes this?
>
> `A5` These appearance changes are more likely due to the overall visual representation and decoding pipeline than to the factorization mechanism itself. In the current setup, FLAM relies on a pretrained tokenizer, so reconstruction errors from the VQ-VAE, caused by limited training data, can lead to color or texture inconsistencies in fine details. This is especially challenging in nuPlan, where scenes are visually rich.
>
> > `Q2` It seems the main benefit comes from Factorizer Temporal Attention. How is a factor defined, and does it need to vary with video complexity?
>
> `A6` We thank the reviewer for this question. In FLAM, a factor refers to one slot in the factorized state representation. Temporal attention does not redefine a factor; it improves temporal consistency by allowing each slot to attend to its own past values across time. Thus, the factor remains the same type of object-level or entity-level component, while temporal attention helps keep it stable across frames.
>
> For more complex videos, one can increase the number of factors $K$ and, if needed, the slot dimension to provide more capacity.

---

> > ### Author Rebuttal · Reviewer_3Lbk · 2026-04-03
> >
> > Thank you for the detailed rebuttal and additional experiments. I appreciate the additional clarification and will increase my score by one point.

---

### Official Review · Reviewer_uLmR · 2026-03-11

**Soundness:** 4
**Presentation:** 3
**Significance:** 3
**Originality:** 3
**Overall Recommendation:** 4
**Confidence:** 4

**Summary:**

This paper proposes FLAM, a factored latent action model that decomposes scenes into independent factors via Slot Attention, each with its own latent action inferred by shared IDM/FDM. This reduces the multi-entity action modeling problem from a joint |A|^K space to per-factor |A| modeling. Experiments on MultiGrid, Procgen (3 games), and nuPlan demonstrate improvements in prediction, representation quality, and policy learning.

**Compliance With Llm Reviewing Policy:**

Affirmed.

**Final Justification:**

Addresses most of my concerns, I will keep the initial positive rating, but evaluations on downstream tasks beyond world model quality remain important.

Additionally, for latent-action evaluations, it would be helpful if the authors could report the FactorVAE score [1], which provides a stronger signal of disentanglement than DCI.

[1] Hyunjik Kim and Andriy Mnih. Disentangling by Factorising. ICML, 2018.

**Key Questions For Authors:**

1. How does FLAM perform when entities strongly interact (e.g., collisions)?
2. Does the Aggregator's use of z_t as query enable trivial "copy-paste" of the current frame, masking actual dynamics prediction quality?
3. Can you provide DCI metrics on Procgen or nuPlan, not just MultiGrid?

**Limitations:**

yes

**Strengths And Weaknesses:**

### Strengths

* The factored decomposition with shared IDM/FDM is well-motivated. End-to-end joint optimization means slots are driven by action independence rather than visual similarity, a meaningful distinction from two-stage approaches like PlaySlot.
* Five datasets spanning grid worlds to real-world driving. Thorough ablations on temporal attention, global coupling, K sensitivity, and β robustness. The correlated entities experiment is clever, confirming action-driven rather than appearance-driven factorization.

### Weaknesses

* The core assumption that factor actions are independent is never tested against strongly interacting entities (collisions, coordination). While IDM/FDM use spatial attention for interaction, latent actions are independently sampled, and coordination may be lost.
* The paper motivates with discrete |A|^K combinatorics but uses continuous VAE-style latent actions. For continuous spaces, the issue is high-dimensional learning difficulty, not discrete combinatorial explosion. The framing should be more precise.
* MultiGrid is a limited (8×8 grid); Procgen has few entities. nuPlan absolute quality (PSNR=19.7) is still far from VQ-VAE ceiling (21.7). Missing validation in truly complex multi-entity scenarios.
* Only 2 environments (Bigfish, Starpilot), no expert upper bound or single BC baseline. No comparison with LAPO or Genie's policy pipeline. Only 2 label sizes tested.
* Slot-Diffusion-based methods ([1] or [2]) are missed in baselines

[1] Wu, Ziyi, et al. "Slotdiffusion: Object-centric generative modeling with diffusion models." Advances in Neural Information Processing Systems 36 (2023): 50932-50958.

[2] Jiang, Jindong, et al. "Object-centric slot diffusion." arXiv preprint arXiv:2303.10834 (2023).

---

> ### Author Rebuttal · Authors · 2026-03-31
>
> We appreciate the reviewer’s time and constructive feedback.
>
> > `W1` / `Q1` FLAM assumes factorized actions, but strong interactions may break this assumption.
>
> `A1` We thank the reviewer for raising this important point. FLAM does not require every entity to map to a separate factor. Instead, it learns an action-driven factorization: when entities act independently, separating them helps, while when their dynamics are strongly correlated, FLAM can group them into the same factor if that improves prediction. This behavior is already partially shown in Section 5.2, where two coordinated agents that always share the same action are consistently assigned to one factor.
>
> To further validate this point, we ran additional experiments on Hallway, a MultiGrid environment where agents must collaborate to pick up keys and open doors.
>
> | Metric | FLAM | AdaWorld |
> |---|---:|---:|
> | PSNR (↑) | 31.5 | 16.5 |
> | SSIM (↑) | 0.956 | 0.708 |
> | LPIPS (↓) | 0.019 | 0.203 |
> | FVD (↓) | 50.5 | 135.0 |
>
> FLAM outperforms AdaWorld. Due to limited computation, we will add other baselines in the next version. These results suggest that factorized latent actions remain effective beyond mostly independent scenes. More broadly, FLAM benefits from factorization when interactions are localized, while retaining the flexibility to group tightly coupled entities when needed.
>
> > `W2` The motivation should reflect continuous latent actions, not only discrete \(|A|^K\) combinatorics.
>
> `A2` We thank the reviewer for this helpful feedback. We agree and will revise the paper to make the motivation more precise. In particular, we will frame FLAM’s benefit as reducing the complexity of modeling joint multi-entity dynamics, rather than only emphasizing discrete action combinations.
>
> > `W3` Current datasets are still limited, and validation on more complex multi-entity scenes is missing.
>
> `A3` We thank the reviewer for this important point. We agree that broader evaluation on more complex multi-entity datasets would strengthen the paper. At the same time, on nuPlan, although the absolute prediction quality remains below the VQ-VAE reconstruction ceiling, FLAM is still the strongest prediction model among the compared methods and consistently the closest to that ceiling. We will revise the discussion to make this interpretation clearer.
>
> > `W4` Downstream evaluation is limited.
>
> `A4` We agree that a more comprehensive downstream evaluation would strengthen the claim. Due to limited computational resources, we were not able to complete these additional experiments in time for the rebuttal. To avoid overstating the result, we will revise the paper to narrow the claim and present the current result as evidence of improved policy learning in the evaluated label-size regime.
>
> > `W5` Diffusion-based slot baselines are missing.
>
> `A5` We thank the reviewer for the suggestion. Our goal is to isolate the effect of factorized latent actions while keeping the overall prediction pipeline as comparable as possible across methods. We therefore prioritize baselines that can share the same pretrained encoder and similar dynamics components, so that improvements are less confounded by major decoder differences.
>
> That said, we agree that combining factorized latent actions with stronger diffusion-based decoders is a promising direction for future work, as discussed in the conclusion.
>
> > `Q2` Does using $z_t$ as query enable trivial copy-paste of the current frame?
>
> `A6` We thank the reviewer for this insightful question. Using $z_t$ as the query helps preserve static details and spatial structure, while the predicted slots model the change from $z_t$ to $z_{t+1}$. This does not permit a trivial copy-paste solution, since the target is $z_{t+1}$, and the model is evaluated with multi-step autoregressive rollouts. If the aggregator merely copied the current frame, errors on moving entities would appear immediately and compound over time.
>
> > `Q3` Can DCI be reported on Procgen or nuPlan?
>
> `A7` We thank the reviewer for the suggestion. Computing DCI requires aligned ground-truth generative variables for each entity, such as positions, so that factor-entity correspondence can be evaluated quantitatively. Procgen and nuPlan do not provide such annotations, making the same DCI evaluation infeasible without additional labeling. We will clarify this point in the revision.

---

> > ### Author Rebuttal · Reviewer_uLmR · 2026-04-01
> >
> > Addresses most of my concerns, but evaluations on downstream tasks beyond world model quality remain important.
> >
> > Additionally, for latent-action evaluations, it would be helpful if the authors could report the FactorVAE score [1], which provides a stronger signal of disentanglement than DCI.
> >
> > [1] Hyunjik Kim and Andriy Mnih. Disentangling by Factorising. ICML, 2018.

---

### Official Review · Reviewer_MEaa · 2026-03-12

**Soundness:** 3
**Presentation:** 3
**Significance:** 3
**Originality:** 3
**Overall Recommendation:** 4
**Confidence:** 4

**Summary:**

This paper studies the problem of learning controllable world models from action-free videos. Prior work on latent action models (LAMs) typically uses an inverse dynamics model to infer a latent action from consecutive frames and a forward dynamics model to predict the next state conditioned on this latent action. However, most existing approaches represent the entire scene with a single monolithic latent action, which becomes problematic in multi-entity environments where multiple objects act independently. To address this limitation, the paper proposes Factored Latent Action Model (FLAM), which introduces factorized latent state and action representations. The method is evaluated on both simulated and real-world multi-entity datasets.

**Compliance With Llm Reviewing Policy:**

Affirmed.

**Key Questions For Authors:**

- Can it works on complex interactions?
- Is pretraining of VQ-VAE necessary? Why?

**Limitations:**

yes

**Strengths And Weaknesses:**

strength

- The paper identifies a well-motivated limitation of existing latent action models: a single latent action must encode all dynamics in a scene. This becomes inefficient when multiple entities move independently.
- The introduction of slot attention sounds and seems to be scalable, despite the experiments are conducted on a small scale.
- Empirical results are convincing. In Table 1 and the figures of the appendix, the results looks great and match the claim of the paper: factorized latent action.

weaknesses

- The method relies on the assumption that scene dynamics can be decomposed into largely independent factors. However, many real-world interactions involve strong coupling between entities (e.g., human-object interaction).
- I wonder that the factorized latent action works on a complex interactions. Since the motivation of latent action is to leverage large-scale video data, it is essential to scale it up.
- Is pretraining of VQ-VAE necessary? Why? Is it possible to train it end-to-end?

---

> ### Author Rebuttal · Authors · 2026-03-31
>
> We appreciate the reviewer’s time, thoughtful comments, and constructive feedback.
>
> > `W1` / `W2` The method relies on the assumption that scene dynamics can be decomposed into largely independent factors. However, many real-world interactions involve strong coupling between entities, for example human-object interaction. Since the motivation of latent actions is to leverage large-scale video data, it is important to understand whether factorized latent actions still work in complex interactive settings.
>
> `A1` We would like to clarify that FLAM does not require every entity to always be assigned to a separate factor. Rather, FLAM learns an action-driven factorization: when entities act independently, separating them is beneficial, but when their dynamics are strongly correlated, FLAM can group them into the same factor if that leads to better prediction. This behavior is already partially supported by our controlled experiment in Section 5.2, where FLAM consistently assigns two correlated agents to a single factor.
>
> In addition, FLAM is not limited to modeling only independent dynamics. In both the IDM and FDM, each factor conditions on all current slots, rather than only its own slot, so cross-entity interactions can still be taken into account. Thus, FLAM can represent interaction effects even when the latent actions are factorized.
>
> To further validate this point, we ran additional experiments on a more interactive environment, Hallway, a MultiGrid setting where agents must collaborate to pick up keys and open doors.
>
> | Metric | FLAM | AdaWorld |
> |---|---:|---:|
> | PSNR (↑) | 31.5 | 16.5 |
> | SSIM (↑) | 0.956 | 0.708 |
> | LPIPS (↓) | 0.019 | 0.203 |
> | FVD (↓) | 50.5 | 135.0 |
>
> As shown above, FLAM outperforms AdaWorld. Due to limited computation, we will add other baselines in the next version. This suggests that factorized latent actions remain effective beyond mostly independent multi-entity scenes. More broadly, FLAM can benefit from factorization when interactions are localized, while still retaining the flexibility to group tightly coupled entities when needed.
>
> > `Q1` Is pretraining the VQ-VAE necessary? Why? Is it possible to train the whole model end-to-end?
>
> `A2` Pretraining the VQ-VAE is not fundamentally necessary. In principle, the whole model can be trained end-to-end. In this work, we separate encoder learning from latent action model learning for two practical reasons. First, pretraining lets the latent action model operate in feature space rather than directly in pixel space, which makes training much more efficient. Second, freezing the encoder helps isolate the effect of the latent action model design itself, rather than conflating it with differences in representation learning. This is also why we use the same pretrained encoder across methods for fair comparison.
>
> That said, we agree that end-to-end training is an interesting direction. It may further improve performance by allowing the tokenizer and latent action model to co-adapt, but it is also more computationally expensive and makes model comparisons less controlled. We will clarify this point in the revision.

---

### Official Review · Reviewer_hA6u · 2026-03-13

**Soundness:** 3
**Presentation:** 3
**Significance:** 3
**Originality:** 3
**Overall Recommendation:** 4
**Confidence:** 3

**Summary:**

This paper proposes the Factored Latent Action Model (FLAM) to address the scalability limits of the LAMs in multi-entity environments. The standard LAM employs a single latent action, which struggles with the combination explosion of simultaneous actions (|A|^K). FLAM decomposes the scene into K independent slots, each with its own latent action and a shared dynamics model. By jointly optimizing slot extraction and dynamics learning, the model achieves an ''action-drive'' decomposition, grouping entities based on their dynamic behaviors rather than visual similarities. Experiments on MultiGrid, Procgen, and nuPlan demonstrate that FLAM outperforms single and two-stage object-centric baselines in terms of prediction quality and downstream policy learning.

**Compliance With Llm Reviewing Policy:**

Affirmed.

**Key Questions For Authors:**

- Could the authors provide a small experiment or discussion on how FLAM performs if the VQ-VAE is trained on a mixture of the Procgen datasets rather than one by one? This would address concerns about the dataset specific nature of the current pipeline.

- Could the author provide a visualization of the ''inactive'' slots when k is over-specified (e.g., k=16 on 4-agent MultiGrid)? Showing that these slots remain ''quiet'' or bind to the background would reinforce the robustness claims.

- How does the factorizer handle moving backgrounds (e.g., camera pan)? Does it require a dedicated background slot, or does the temporal attention naturally handle this as a ''shared'' action?

**Limitations:**

Yes

**Strengths And Weaknesses:**

Strengths:

+ This paper is easy to follow.

+ Decomposing the action space from |A|^K to K×|A| is a theoretically reasonable and practically effective approach for handling multi-agent scenarios.

+ The joint training paradigm (Section 4.2) is a significant advancement compared to the two-stage object-centered models. Figure 7 effectively demonstrates that FLAM is capable of grouping entities with the same dynamic characteristics but different appearances.

+ FLAM consistently outperforms modern benchmark models such as Genie and AdaWorld in simulated data and real-world (nuPlan) data, as evidenced by metrics such as PSNR, SSIM, and FVD.

Concerns:

- For each dataset, a separate VQ-VAE is trained and frozen. This limits the "generalization" potential of the world model. If the encoder is not shared or pre-trained on a broader scale, this decomposition heavily relies on the quality of the initial feature extraction, and in more visually diverse "real-world" settings, this quality may not be maintained. Therefore, this may not be a universal model that can exhibit robust performance across various scenarios.

- In nuPlan, this model uses 16 factors to identify vehicles and pedestrians. However, in actual scenarios with high background complexity (such as trees, changes in light, and weather conditions), it is not clear whether this factor identifier will waste resources on environmental noise rather than controllable entities, or whether the "background slot" mechanism is sufficient. Should distinctions be introduced on these factors to better focus attention on the objects that should be identified?

- Although the prediction accuracy is very high, the visualization results (in Figure 14) indicate that the model still has difficulties in handling high-frequency details. Compared with modern diffusion-based decoders, the current transformer-based aggregators may be a bottleneck.

- The Benchmark test only takes into account the player's actions, but this approach makes the comparison results somewhat unfair because FLAM actually discovers potential "labels" for all entities. It would be helpful to briefly explain why the "real conditions" of all entities were not taken as the theoretical upper limit value.

- The paper mentioned the robustness of the KL coefficient β, but the ''slot attention'' algorithm is extremely sensitive to the initialization and the number of slots (k). Further discussions on the stability of the factorizer during the joint training stage would be more helpful for the readers.

- While the factorizer handles multiple entities, there is limited discussion on how it manages complex, non-controllable background movement (e.g., trees or weather in nuPlan).

---

> ### Author Rebuttal · Authors · 2026-03-31
>
> > `W1` / `Q1` Separate VQ-VAE per dataset limits generalization
>
> `A1` We thank the reviewer for this important point. We agree that sharing the encoder across visually diverse domains is an important direction toward broader generalization, and we already list this as a limitation and future direction in the paper.
>
> Following the reviewer’s suggestion, we additionally trained a shared VQ-VAE on a mixture of the Procgen datasets and evaluated FLAM and AdaWorld on top of that shared encoder. We observe slightly lower performance than with individually trained encoders. However, FLAM still outperforms the AdaWorld baseline. Due to limited computation, we will add other baselines in the next version of the paper. This result suggests that FLAM’s gains do not rely on a strictly dataset-specific tokenizer.
>
> **Bigfish**
>
> | Metric | FLAM (shared encoder) | AdaWorld (shared encoder) | FLAM (separate encoder) | AdaWorld (separate encoder) |
> |---|---:|---:|---:|---:|
> | PSNR (↑) | 30.8 | 30.3 | 32.4 | 31.8 |
> | SSIM (↑) | 0.983 | 0.982 | 0.991 | 0.984 |
> | LPIPS (↓) | 0.011 | 0.012 | 0.006 | 0.009 |
> | FVD (↓) | 63.6 | 63.3 | 27.2 | 40.3 |
>
> **Starpilot**
>
> | Metric | FLAM (shared encoder) | AdaWorld (shared encoder) | FLAM (separate encoder) | AdaWorld (separate encoder) |
> |---|---:|---:|---:|---:|
> | PSNR (↑) | 29.3 | 27.8 | 30.7 | 29.3 |
> | SSIM (↑) | 0.971 | 0.965 | 0.978 | 0.971 |
> | LPIPS (↓) | 0.015 | 0.023 | 0.008 | 0.014 |
> | FVD (↓) | 73.4 | 113.4 | 40.7 | 69.3 |
>
> > `W2` / `W6` Background complexity and non-controllable motion in nuPlan
>
> `A2` We thank the reviewer for this thoughtful question. We view this issue in two parts.
>
> First, in latent action models, the latent action is intended to capture information that is not predictable from the current state alone. Therefore, if background content is static, or if its motion is largely predictable from history, the model does not need to devote much latent-action capacity to it.
>
> Second, if the background contains unpredictable motion, this is a harder case not only for FLAM but for latent action models more generally. Since the training objective optimizes prediction error uniformly over the observation, without semantic supervision about what is controllable or task-relevant, the model may allocate capacity to such background changes as well. In that sense, the current formulation does not explicitly distinguish controllable entities from unpredictable but non-controllable motion.
>
> In practice, a factor can bind to background regions or shared scene structure, but this behavior is emergent rather than explicitly enforced. We agree that adding stronger priors or supervision, for example objectness, controllability, or task relevance, would be a promising direction for improving focus in visually complex real-world scenes. We will add this discussion to the paper.
>
> > `W3` High-frequency details in Figure 13
>
> `A3` We agree that the visualizations in Figure 13 suggest that high-frequency image details are still not fully captured. We believe this is at least partly due to the current latent-space prediction and decoding pipeline, rather than the factorization idea itself.
>
> Our main contribution is the factorized latent action formulation for multi-entity dynamics. Exploring stronger decoders, such as diffusion-based or flow-matching decoders, is a natural next step and could further improve visual fidelity, as discussed in the conclusion.
>
> > `W4` Procgen comparison only uses player actions
>
> `A4` We agree that using ground-truth actions for all entities would be the ideal upper bound. Unfortunately, in Procgen only the player action is available, so such an upper bound cannot be constructed from the dataset.
>
> > `W5` / `Q2` Stability of slot attention during joint training
>
> `A5` We thank the reviewer for raising this point. We agree that factorizer stability is important in jointly trained slot-based models.
>
> Regarding the number of slots, we already include an ablation over $K \in [2, 4, 8, 16]$ in Figure 7. The results show that performance degrades when \(K\) is smaller than the number of independently acting entities, but becomes stable once \(K\) is sufficiently large. This suggests that FLAM is reasonably robust to over-specifying \(K\).
>
> Regarding initialization, in our implementation we use separate learnable slot embeddings, which we found to be the most stable choice in preliminary experiments compared with Gaussian-based initializations. We will add a brief clarification on this implementation choice in the revision. In addition, our factorizer includes causal temporal attention, which is specifically designed to improve slot consistency across time. As shown in our ablations, removing this component substantially hurts both prediction quality and factor-entity correspondence, indicating that temporal consistency is important for stable joint training.

---

> > ### Author Rebuttal · Reviewer_hA6u · 2026-04-04
> >
> > The rebuttal thoroughly addressesed my concerns, I will therefore recommend acceptance.

---

### Decision · Program_Chairs · 2026-04-30

**Decision:**

Accept (regular)

**Comment:**

The paper introduces FLAM, a world model learning latent actions from action-free video. Reviewers agree that the paper is well motivated and achieves good results against strong baselines. Given the positive consensus, I recommend acceptance.